# Phenotypic and molecular evolution across 10,000 generations in laboratory budding yeast populations

**Milo S Johnson**[1,2,3]*, **Shreyas Gopalakrishnan**[1,2,3,4], **Juhee Goyal**[1,5],
**Megan E Dillingham**[2,6], **Christopher W Bakerlee**[1,2,3,4], **Parris T Humphrey**[1,2,3],
**Tanush Jagdish**[1,2,3,6], **Elizabeth R Jerison**[1,7,8], **Katya Kosheleva**[1,7],
**Katherine R Lawrence**[1,2,3,9], **Jiseon Min**[1,2,3,4,5], **Alief Moulana**[1], **Angela M Phillips**[1],
**Julia C Piper**[1,10], **Ramya Purkanti**[1,11], **Artur Rego-Costa**[1], **Michael J McDonald**[1,12],
**Alex N Nguyen Ba**[1,2,3,7,13], **Michael M Desai**[1,2,3,7]*

[1]Department of Organismic and Evolutionary Biology, Harvard University, Cambridge, United States; [2]Quantitative Biology Initiative, Harvard University, Cambridge, United States; [3]NSF-Simons Center for Mathematical and Statistical Analysis of Biology, Harvard University, Cambridge, United States; [4]Department of Molecular and Cellular Biology, Harvard University, Cambridge, United States; [5]John A Paulson School of Engineering and Applied Sciences, Harvard University, Cambridge, United States; [6]Graduate Program in Systems, Synthetic, and Quantitative Biology, Harvard University, Cambridge, United States; [7]Department of Physics, Harvard University, Cambridge, United States; [8]Department of Applied Physics, Stanford University, Stanford, United States; [9]Department of Physics, Massachusetts Institute of Technology, Cambridge, United States; [10]AeroLabs, Aeronaut Brewing Co, Somerville, United States; [11]The Max Planck Institute of Molecular Cell Biology and Genetics, Dresden, Germany; [12]School of Biological Sciences, Monash University, Victoria, Monash, Australia; [13]Department of Cell and Systems Biology, University of Toronto, Toronto, Canada

*For correspondence:
milo.s.johnson.13@gmail.com
(MSJ);
mdesai@oeb.harvard.edu (MMD)

Competing interest: See
page 23

Reviewing editor: Kevin J
Verstrepen, VIB-KU Leuven
Center for Microbiology,
Belgium

**Abstract** Laboratory experimental evolution provides a window into the details of the evolutionary process. To investigate the consequences of long-term adaptation, we evolved 205 *Saccharomyces cerevisiae* populations (124 haploid and 81 diploid) for ~10,000 generations in three environments. We measured the dynamics of fitness changes over time, finding repeatable patterns of declining adaptability. Sequencing revealed that this phenotypic adaptation is coupled with a steady accumulation of mutations, widespread genetic parallelism, and historical contingency. In contrast to long-term evolution in *E. coli*, we do not observe long-term coexistence or populations with highly elevated mutation rates. We find that evolution in diploid populations involves both fixation of heterozygous mutations and frequent loss-of-heterozygosity events. Together, these results help distinguish aspects of evolutionary dynamics that are likely to be general features of adaptation across many systems from those that are specific to individual organisms and environmental conditions.

## Introduction

As human health is increasingly threatened by emerging pathogens, multidrug-resistant infections, and therapy-evading cancer cells, our understanding of the dynamics and predictability of evolution is of growing importance. Yet predicting the course of evolution is difficult, since it is driven by a

complex combination of deterministic and stochastic forces. On the one hand, beneficial mutations that establish within a population often rise to fixation at rates nearly perfectly predicted by decades-old theory. On the other hand, random forces such as mutation, genetic drift, and recombination ensure an enduring role for chance and contingency. To understand evolution, we must appreciate the interactions between these deterministic and stochastic components.

While there is extensive theoretical work analyzing how the interplay between these factors affects the rate, predictability, and molecular basis of evolution, empirical evidence remains relatively limited. In large part, this stems from a basic difficulty: we cannot easily characterize the predictability of evolution using observational studies of natural populations, because we cannot replicate evolutionary history. In addition, the inferences we can make from extant populations and the fossil record are limited by a lack of complete data.

To circumvent these difficulties, scientists have turned to laboratory evolution experiments, primarily in microbial populations. These provide a simple model system in which researchers can maintain many replicate populations for hundreds or thousands of generations, in a setting where the environment and other relevant parameters (e.g. population size) can be precisely controlled and manipulated. By conducting phenotypic and sequencing studies of the resulting evolved lines, we can observe evolution in action, and ask whether specific phenotypic and genotypic outcomes are predictable.

Over the last several decades, a few consistent results have emerged from these types of experiments (reviewed in *Kassen, 2014*). As populations evolve in a constant environment, they gain fitness along a fairly predictable trajectory, following a pattern of declining adaptability in which the rate of fitness increase slows as populations adapt (*Couce and Tenaillon, 2015*; *Kryazhimskiy et al., 2014*; *Wiser et al., 2013*). Meanwhile, the rate of molecular evolution remains roughly constant (*Barrick et al., 2009*; *Good et al., 2017*; *Tenaillon et al., 2016*). Mutations are rarely predictable at the nucleotide level but often moderately predictable at higher levels: mutations in certain genes or pathways are repeatedly fixed across replicate populations (*Bailey et al., 2015*; *Kryazhimskiy et al., 2014*; *Tenaillon et al., 2012*; *Tenaillon et al., 2016*). Phenotypes not under direct selection change less predictably than fitness in the evolution environment, but sometimes still exhibit some correlation with level of adaptation in the evolution environment (*Jerison et al., 2020*; *Leiby and Marx, 2014*; *Ostrowski et al., 2005*).

Most of these microbial and viral evolution experiments, as well as those in multicellular eukaryotes such as *C elegans* and *Drosophila melanogaster*, involve at most about 1000 generations of adaptation to a novel environment. This makes them well suited to studying the initial dynamics of adaptation, where a population encounters a novel environment and rapidly acquires beneficial mutations as it evolves in response to this new challenge. However, it is unclear how far we can extrapolate findings from this type of study. Will evolutionary dynamics remain similar over longer timescales? Or will the evolutionary dynamics change in qualitative ways once a population has had thousands of generations to become well-adapted to the laboratory environment?

The experiment best equipped to answer this question is the Long-Term Evolution Experiment (LTEE) conducted by Richard Lenski and collaborators. For over 30 years and 70,000 generations (reviewed in *Lenski, 2017*), the Lenski lab has propagated 12 *Escherichia coli* populations in minimal media by batch culture. The LTEE has led to numerous insights into evolutionary dynamics over both short and long timescales, and has also provided many examples of interesting phenomena such as contingency (*Blount et al., 2012*; *Good et al., 2017*), the spontaneous emergence of quasi-stable coexistence (*Good et al., 2017*; *Plucain et al., 2014*; *Rozen and Lenski, 2000*), and evolution of mutation rates (*Sniegowski et al., 1997*; *Wielgoss et al., 2013*). The LTEE is unique among microbial evolution experiments in its long timescale, and provides an important look at evolution well beyond the initial rapid adaptation of a population to a novel laboratory environment. However, it is limited by its specificity: it involves 12 replicate populations, each founded from a single *E. coli* strain, all evolving in the same constant environment. It thus remains unclear which of the broad conclusions drawn from this experiment will be generalizable to other organisms and environments. Would we draw similar conclusions when other species are allowed to evolve in other environments for long periods of time?

While no other laboratory evolution experiments match the LTEE in timescale, a few have extended beyond the ~1000 generations of most other experiments. For example, *Behringer et al., 2018* evolved *E. coli* populations in tubes for up to 10,000 generations and found that they

repeatedly evolved a biofilm phenotype and stable coexisting subpopulations. *Fisher et al., 2018* evolved laboratory populations of the budding yeast *S. cerevisiae* for 4000 generations, finding that as in *E. coli*, these populations gain fitness along predictable trajectories characterized by declining adaptability. This experiment, along with *Marad et al., 2018*, also studied the relationship between ploidy and adaptation, finding that in general diploids adapt more slowly than haploids. Slower adaptation in diploids has been observed in yeast evolution experiments in a variety of environments and appears to be caused by the reduced efficacy of selection on recessive or partially recessive beneficial mutations in diploids (*Zeyl et al., 2003*; *Gerstein et al., 2011*). While these experiments provide an important first look into long-term adaptation in yeast and *E. coli*, they all involve relatively limited whole-population sequencing, and none have provided data on the dynamics of molecular evolution in both haploid and diploid populations over many thousands of generations.

To fill this gap, we established a long-term evolution experiment in the spirit of the LTEE, with a total of 205 budding yeast populations (split between haploids and diploids) evolving in three different laboratory environments. In this paper, we describe the first 10,000 generations of this experiment. We find that some aspects of evolution in our system are broadly consistent with the conclusions of the LTEE and other long-term evolution experiments. For example, the dynamics of fitness increase are largely repeatable between replicate lines and show a pattern of declining adaptability over time even while the rate of molecular evolution remains relatively constant. However, there are also key differences: we find no evidence of stably coexisting lineages or widespread evolution of mutator phenotypes. As the first laboratory evolution of this length in a eukaryotic system, our study provides an important test of the generality of conclusions from earlier work (primarily the LTEE), as well as a novel opportunity to observe evolutionary dynamics over long timescales across many replicate populations in multiple environmental conditions.

## Results

We founded 45 haploid mating type a (*MATa*), eight mating type α (*MATα*), and 37 diploid *S. cerevisiae* populations in each of three evolution environments (90 populations per environment, for a total of 270 independent lines; see *Figure 1*). Each population was founded from a single independent colony of the corresponding ancestral W303 *MATa*, *MATα*, or diploid strain (see Materials and methods for details). We then propagated each population in batch culture in one well of an unshaken

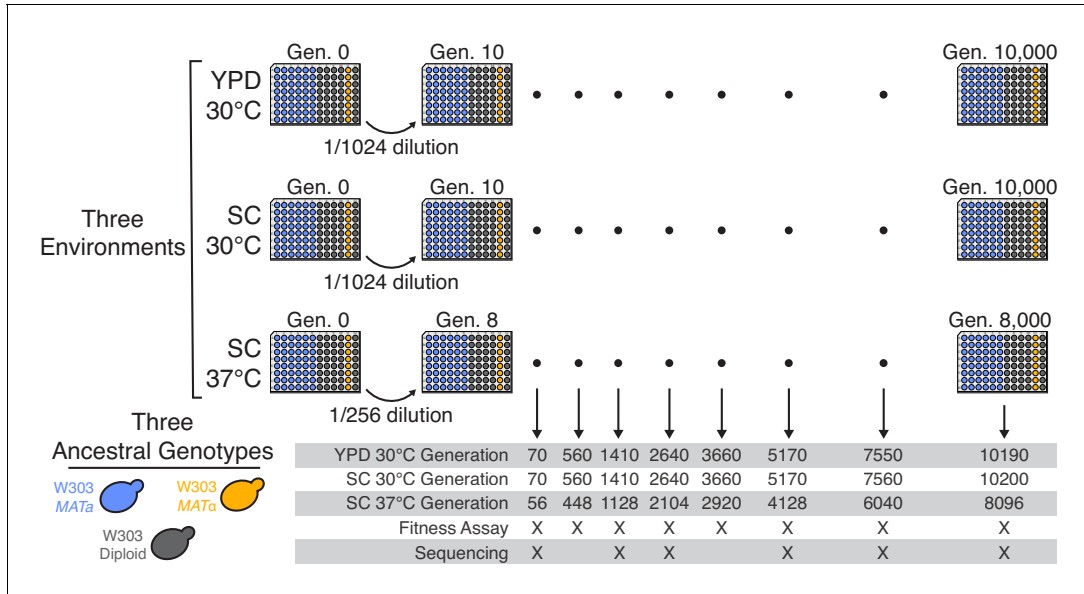

**Figure 1.** Experimental design. We propagated budding yeast lines in 96-well microplates in one of three environmental conditions, using a daily dilution protocol as shown at top. Each population was founded by a single clone of one of three ancestral genotypes (a haploid *MATa*, a haploid *MATα*, and a diploid, all derived from the W303 strain background). On a weekly basis, we froze all populations in glycerol at −80˚C for long-term storage. The frozen timepoints used for the analyses in this paper are indicated at bottom.

96-well microplate in the appropriate environment (YPD at 30℃, SC at 30℃, and SC at 37℃), with daily 1:2^10 dilutions for the 30℃ environments, and 1:2^8 dilutions for the 37℃ environment. We froze glycerol stocks of each population every week (corresponding to every 70 generations in the 30℃ environments, and every 56 generations in the 37℃ environment), creating a frozen fossil record for future analysis. A total of 65 populations were lost during the first 10,000 generations of evolution due to contamination, evaporation, or pipetting errors (see Materials and methods for details; *Supplementary file 1*), leaving us with 205 populations.

## Fitness changes during evolution

To measure changes in fitness over time, we unfroze populations from eight timepoints in each of the 205 evolved populations (see *Figure 1*) and conducted competitive fitness assays against a fluorescently labeled reference strain (see Materials and methods for details). In *Figure 2*, we show the resulting fitness trajectories in each population. We find that in most cases, including almost all haploid populations, these trajectories tell a familiar story of declining adaptability: populations predictably increase in fitness rapidly in the first few hundred generations, and then adapt more slowly as time progresses (*Figure 2—figure supplement 1A*). We find a different pattern in some diploid populations in SC 30℃, where an initial slower period of fitness gain is succeeded by a significant rapid increase in fitness. We also find that a few populations (indicated by asterisks in *Figure 2*) experience dramatic increases in fitness, and subsequently remain at higher fitness than other populations for the duration of the experiment. As we describe in more detail below, these events are caused by a specific mutation in the adenine biosynthesis pathway (see 'ADE pathway mutations' section).

On average, our haploid populations gained more fitness over the course of evolution than diploids ($p<0.02$, Mann-Whitney U test), consistent with prior work (*Fisher et al., 2018*; *Marad et al., 2018*; *Zeyl et al., 2003*; *Gerstein et al., 2011*). This effect could be entirely due to reduced accessibility of recessive or partially recessive beneficial mutations in diploids (as these previous studies propose). However, we note that this effect is also consistent with declining adaptability: the diploids have a higher ancestral fitness than the haploids in all three environments (*Figure 2*). We also observe more rapid evolution of *MATa* haploids compared to *MATα* haploids in SC 30℃, where the *MATa* haploids start at lower fitness (*Figure 2—figure supplement 1B*; $p<0.02$, Mann-Whitney U test). Together, these results are consistent with a picture in which the pattern of declining adaptability as a function of fitness applies not only along the course of a fitness trajectory for one

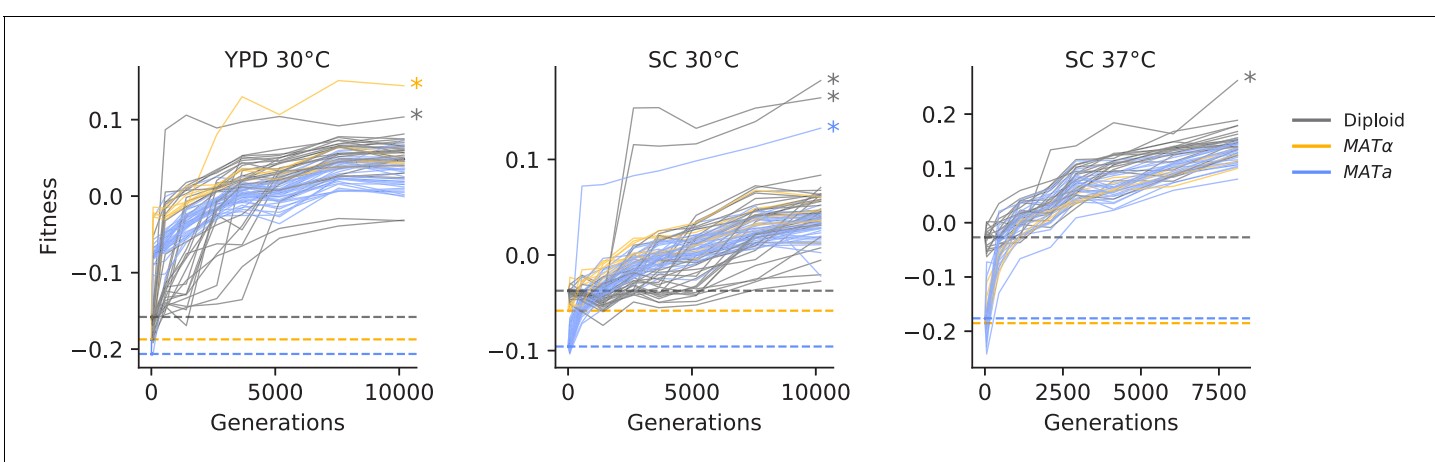

**Figure 2.** Fitness changes during evolution. Competitive fitness is plotted relative to a reference strain in each environment. Inferred ancestral fitness is indicated by horizontal lines and colored by strain. Populations with premature stop-codon reversion mutations in ADE2 are indicated by asterisks. Correlations between replicate fitness measurements are shown in *Figure 2—figure supplement 2*.

The online version of this article includes the following figure supplement(s) for figure 2:

**Figure supplement 1.** Declining adaptability.

**Figure supplement 2.** Correlations between absolute fitness measured in replicate competitions with a fluorescent reference.

population, but also between ancestral strains of different ploidy and mating type (*Figure 2—figure supplement 1*).

## Molecular evolution

At six of the timepoints used for fitness assays (*Figure 1*), we also performed whole-population, whole-genome sequencing in 90 focal populations (12 *MATa*, 12 diploid, and 6 *MATα* from each environment). After aligning sequencing reads and calling variants, we use observed allele counts across multiple timepoints to filter out sequencing and alignment errors and identify a set of mutations present in each evolving population (Materials and methods). At each sequenced timepoint, we call mutations fixed if they are at greater than or equal to 40% frequency (diploids) or 90% frequency (haploids) and do not drop below these thresholds at a later timepoint. We additionally call loss of heterozygosity in mutations in diploids using the criteria for fixation in haploids (90% threshold).

Our data shows that mutations fix steadily through time across all sequenced populations (*Figure 3*). While we would need more sequenced timepoints to fully observe the frequency trajectories of mutations in these populations, we can see a few patterns from our temporally sparse sequencing (*Figure 3A*, *Figure 3—figure supplement 1–9*). We frequently observe clonal interference in which groups of mutations rise to high frequency and then plummet to extinction, outcompeted by another group. All populations fix mutations throughout the experiment; we find no evidence for the emergence of stably coexisting lineages within any of our populations (*Figure 3B*, *Figure 3—figure supplement 10*). Denser sequencing through time would be required to determine whether any populations exhibit shorter periods of semistable coexistence (e.g. as seen by *Frenkel et al., 2015*). It is also possible we are missing coexistence of haplotypes at very low frequency ($\leq$5%), which sequencing may not be able to detect. However, our results rule out long-term coexistence of multiple lineages at substantial frequencies like that observed after 10,000 generations of evolution in the LTEE or *Behringer et al., 2018*.

We find that the rate of mutation accumulation in the *MATa* populations is consistently higher than in *MATα* or diploid populations (*Figure 3B*). This is likely due to a higher mutation rate in our *MATa* ancestor. Consistent with this hypothesis, we find that *MATa* populations have a lower ratio of nonsynonymous to synonymous mutations than *MATα* or diploid populations in all three environments, as expected if a higher mutation rate leads to an increase in hitchhiking (although we note that this comparison is only significant in SC 37°C; p<0.01, Mann-Whitney U Test, *Figure 4A*). We identified a putative causal mutation in TSA1 in our *MATa* ancestor; this mutation is absent in our *MATα* ancestor and heterozygous in our diploid ancestor. We confirmed that the TSA1 mutation increases mutation rate in a BY strain background (*Figure 4—figure supplement 1*).

Overall, we find that dN/dS ratios for fixed mutations in our populations are near one (*Figure 4A*), suggesting that selection in favor of beneficial (and presumably typically nonsynonymous) mutations is balanced by hitchhiking of neutral mutations and purifying selection against deleterious mutations. The relative prevalence of different types of fixed mutations across strains and environments are similar, with roughly 45–50% missense mutations, 40–45% synonymous and noncoding mutations, and 5–10% nonsense and indel mutations (*Figure 4B*). While there is variation between populations in the number of mutations accumulated, we do not observe any sudden increases in the rate of mutation accumulation (*Figure 3B*). This stands in contrast to the LTEE, where mutator alleles sweep to fixation and dramatically increase the mutation rate in 6 of 12 replicate populations (four of which are apparent in sequencing data after 10,000 generations [*Good et al., 2017*]). We do observe one potential mutator event: P1E11, an *MATα* population evolved in YPD 30°C has an unusually large number of indel mutations, likely due to a mutation in the mismatch repair protein MSH3 that hitchhiked to fixation with an indel mutation in GPB2 (*Figure 4—figure supplement 2*). However, the elevation in mutation rate in this population remains relatively modest. While we do observe mutations in mutator-associated genes such as MSH3 in other populations, we do not observe clear differences in mutation-type distribution or rate of mutation accumulation in these populations, suggesting that these mutations lead to at most subtle changes in mutation rate (stacked mutation type plots for each population are shown in *Figure 4—figure supplements 3–14*). Further work will be needed to characterize more subtle variation in mutation rate in each of these populations.

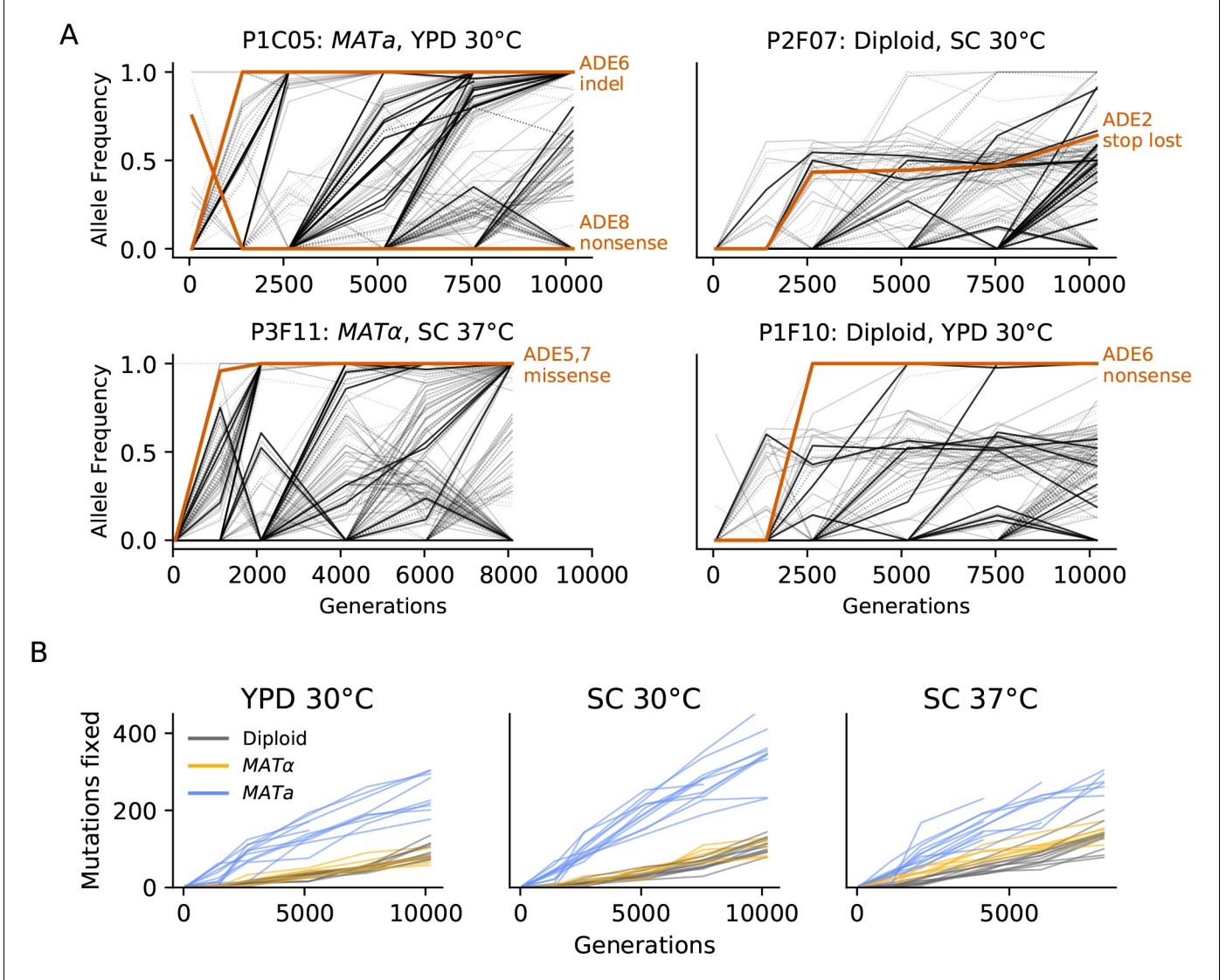

**Figure 3.** Dynamics of molecular evolution. (**A**) Allele frequencies over time in four example populations. Nonsynonymous mutations in 'multi-hit' genes are solid black lines (see 'Parallelism' section below), nonsynonymous mutations in the adenine biosynthesis pathway are colored orange and labeled, other nonsynonymous mutations are thin gray lines, and synonymous mutations are dotted lines. (**B**) Number of fixed mutations over time in each population. Timepoints with average coverage less than 10 (for haploids) or 20 (for diploids) are not plotted.

The online version of this article includes the following figure supplement(s) for figure 3:

**Figure supplement 1.** Allele frequencies over time in all focal diploid populations in YPD 30˚C.
**Figure supplement 2.** Allele frequencies over time in all focal *MATa* populations in YPD 30˚C.
**Figure supplement 3.** Allele frequencies over time in all focal *MATα* populations in YPD 30˚C.
**Figure supplement 4.** Allele frequencies over time in all focal diploid populations in SC 30˚C.
**Figure supplement 5.** Allele frequencies over time in all focal *MATa* populations in SC 30˚C.
**Figure supplement 6.** Allele frequencies over time in all focal *MATα* populations in SC 30˚C.
**Figure supplement 7.** Allele frequencies over time in all focal diploid populations in SC 37˚C.
**Figure supplement 8.** Allele frequencies over time in all focal *MATa* populations in SC 37˚C.
**Figure supplement 9.** Allele frequencies over time in all focal *MATα* populations in SC 37˚C.
**Figure supplement 10.** No evidence of coexistence.
**Figure supplement 11.** Copy number variation in the ribosomal DNA array and *CUP1* array, determined from sequencing coverage data.

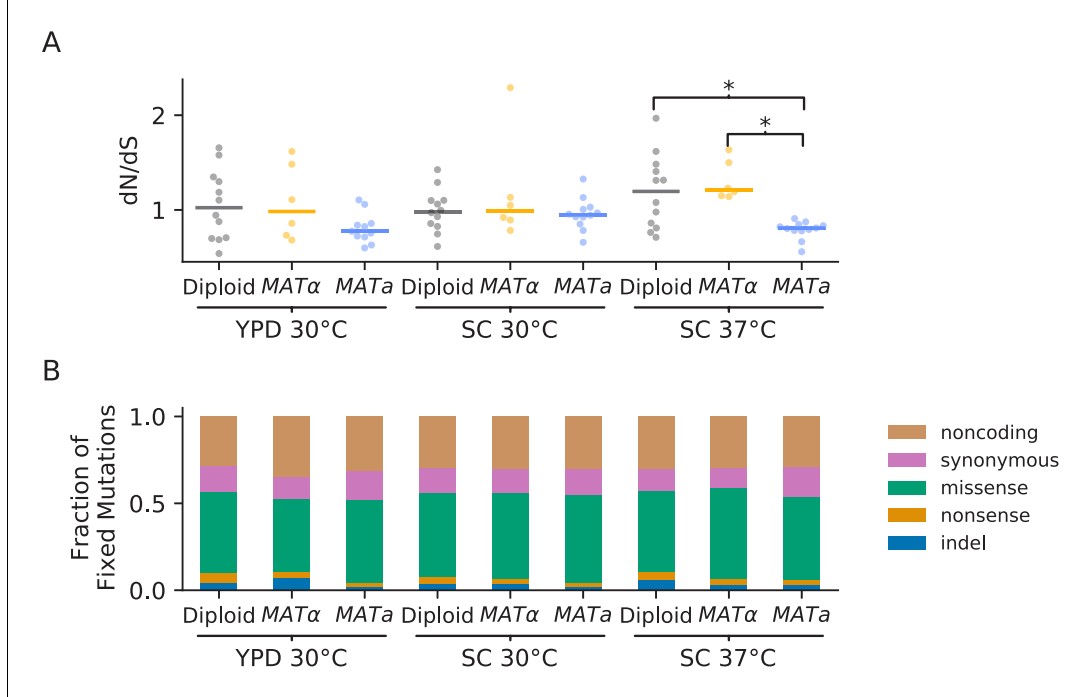

**Figure 4.** Types of mutations. (**A**) Swarm plot of dN/dS (ratio of nonsynonymous / synonymous fixations by the final timepoint, scaled by the ratio of possible nonsynonymous / synonymous mutations across the genome) for each environment-strain combination. Each point represents one population and the horizontal line represents the median. Asterisks indicate significant differences (p<0.01, Mann-Whitney U test) between strains in the same environment. (**B**) Breakdown of mutation types for all mutations fixed by the final timepoint, in all populations corresponding to each environment-strain combination.

The online version of this article includes the following figure supplement(s) for figure 4:

**Figure supplement 1.** Confirmation that the TSA1 mutation increases mutation rate.

**Figure supplement 2.** Population P1E11, a putative mutator.

**Figure supplement 3.** Stacked plot of heterozygous or homozygous fixed mutation types over time in all focal diploid populations in YPD 30°C.

**Figure supplement 4.** Stacked plot of homozygous-only (lost heterozygosity) fixed mutation types over time in all focal diploid populations in YPD 30°C.

**Figure supplement 5.** Stacked plot of fixed mutation types over time in all focal *MATa* populations in YPD 30°C.

**Figure supplement 6.** Stacked plot of fixed mutation types over time in all focal *MATα* populations in YPD 30°C.

**Figure supplement 7.** Stacked plot of heterozygous or homozygous fixed mutation types over time in all focal diploid populations in SC 30°C.

**Figure supplement 8.** Stacked plot of homozygous-only (lost heterozygosity) fixed mutation types over time in all focal diploid populations in SC 30°C.

**Figure supplement 9.** Stacked plot of fixed mutation types over time in all focal *MATa* populations in SC 30°C.

**Figure supplement 10.** Stacked plot of fixed mutation types over time in all focal *MATα* populations in SC 30°C.

**Figure supplement 11.** Stacked plot of heterozygous or homozygous fixed mutation types over time in all focal diploid populations in SC 37°C.

**Figure supplement 12.** Stacked plot of homozygous-only (lost heterozygosity) fixed mutation types over time in all focal diploid populations in SC 37°C.

**Figure supplement 13.** Stacked plot of fixed mutation types over time in all focal *MATa* populations in SC 37°C.

**Figure supplement 14.** Stacked plot of fixed mutation types over time in all focal *MATα* populations in SC 37°C.

## Parallelism

Next, we examined whether mutations in certain genes are fixed more frequently than we would expect by chance. We define a 'hit' as a nonsynonymous mutation that is fixed by the final timepoint, and define the multiplicity of a gene as the number of hits in that gene across all sequenced populations, divided by its relative target size (*Good et al., 2017*). As in many other laboratory evolution experiments, we observe an excess of high multiplicity genes in our data, relative to a null in which mutations are fixed randomly across all open-reading frames (*Figure 5A*).

To understand the functional basis of this parallelism, we focus on multi-hit genes, defined as those with hits in six or more populations. These multi-hit genes (*Figure 6*) are enriched for several gene ontology (GO) terms (*Supplementary file 4*), indicating parallelism at the level of biosynthetic

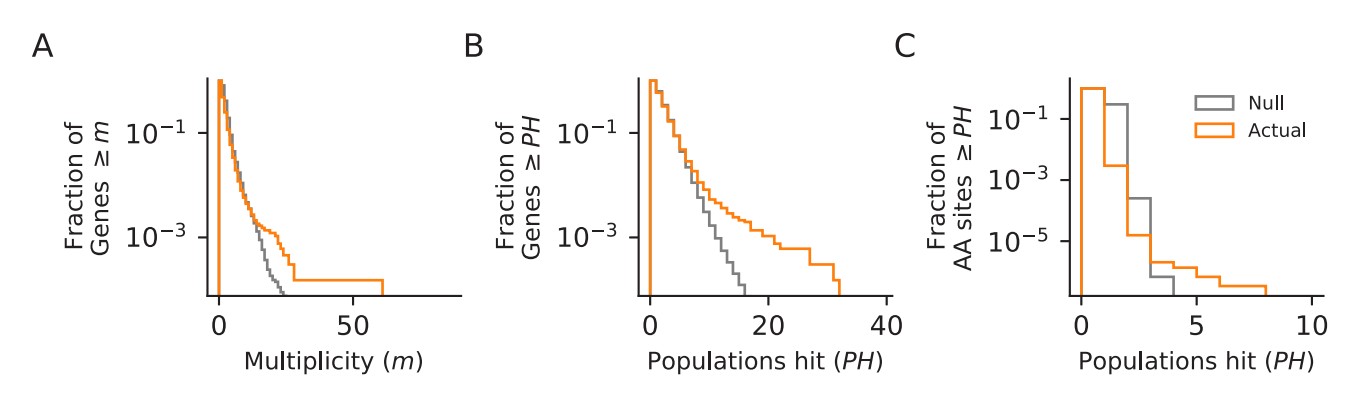

**Figure 5.** Parallelism. Comparison between null and actual distributions of (**A**) the fraction of genes with multiplicity $\geq m$ (see Materials and methods), (**B**) the fraction of genes with hits in $\geq PH$ populations, and (**C**) the fraction of amino acid sites with hits in $\geq PH$ populations (those with $PH \geq 3$ are listed in **Supplementary file 4**). For all three plots, the null distribution (shown in gray) is obtained by simulating random hits to genes, taking into account the number of hits in each population in our data and the relative length of each gene.

and signaling pathways. In **Figure 6**, we show all genes with hits in ten or more populations, and highlight several key functional groups (adenine biosynthesis, sterility, and negative regulators of the Ras pathway; see **Figure 6—figure supplements 1–3** for analogous figures for all other multi-hit genes). Mutations in the latter two functional groups are commonly observed in yeast evolution experiments, and have been shown to be beneficial in similar environments (**Rojas Echenique et al., 2019**; **Kryazhimskiy et al., 2014**; **Lang et al., 2013**; **Venkataram et al., 2016**). The mutations in adenine biosynthesis, by contrast, reflect the particular genotype of our ancestral strains; we discuss these further below.

We next asked whether some multi-hit genes are more likely to fix mutations in particular strain backgrounds or environments. We find that most multi-hit genes have mutations distributed across both haploid mating types, diploids, and all three environmental conditions, indicating that these mutations are presumably beneficial in all these contexts. However, we do find several mutations that are either strain or environment specific ('Effect' column in **Figure 6**; **Figure 6—figure supplement 4**, **Supplementary file 4**). For example, mutations in SRS2 and LCB3 are fixed more often in SC 37°C, while mutations in CCW12 are fixed more in diploids.

To investigate the impact of the mutations in multi-hit genes on protein function, we used SnpEff (**Cingolani et al., 2012**) to predict the impact of each mutation. In **Figure 6**, we show the fraction of mutations in each multi-hit gene that were annotated as 'High Impact'. Because most of these high-impact mutations are nonsense or frameshift mutations, they are very likely to lead to loss of function of the associated gene, as are some fraction of the 'Moderate Impact' mutations (e.g. some missense mutations or in-frame deletions). We find that many of our multi-hit genes have a large percentage of high-impact mutations, suggesting that selection acts in favor of loss-of-function of the corresponding genes, consistent with many earlier laboratory evolution experiments (**Murray, 2020**). However, this is not universal: a few genes with 10 or more hits have no high-impact mutations fixed, and several of these genes are essential (**Figure 6**). This suggests that selection in these genes may be instead for change- or gain-of-function.

## ADE pathway mutations

The founding genotype of all the populations used in this experiment is derived from the W303 strain background, which has a premature stop codon in the ADE2 gene (*ade2-1*). This disrupts the adenine biosynthetic pathway, which is likely deleterious because adenine depletion can limit growth even in rich media (**Kokina et al., 2014**). In addition, loss-of-function of ADE2 causes buildup of a toxic intermediate, phospho-ribosylaminoimidazole (AIR), which is converted to a visible red pigment that accumulates in the vacuole (**Kokina et al., 2014**; **Sharma et al., 2003**; **Figure 7A**). This means that loss-of-function mutations upstream in this pathway, which are deleterious when ADE2 is

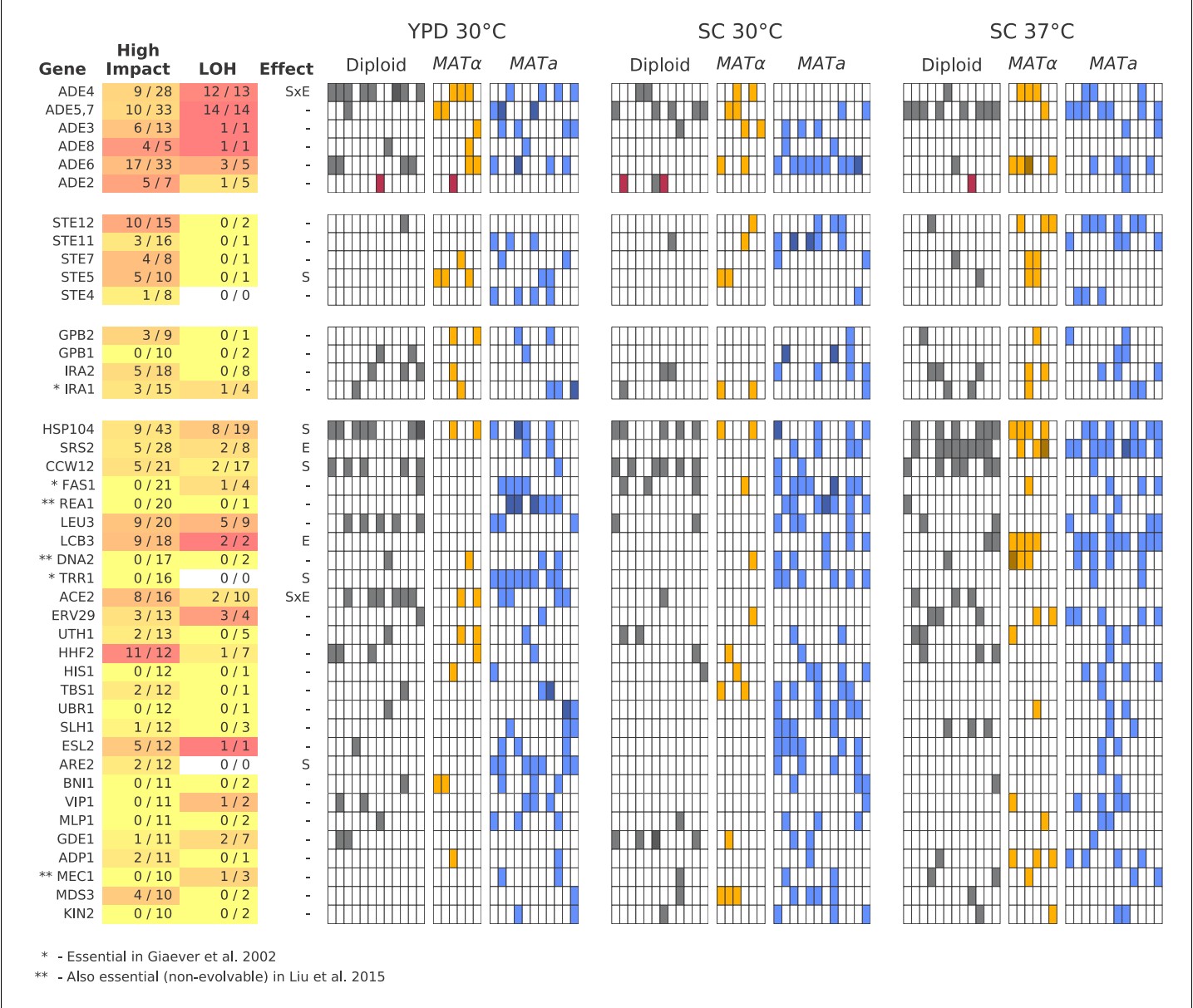

**Figure 6.** Multi-hit genes. Each row represents a gene. The first three blocks are groups of genes identified from gene-ontology enrichment analysis of multi-hit genes (from top to bottom: adenine biosynthesis, sterility, and negative regulation of the Ras pathway). The bottom block is all other genes with hits in at least 10 populations. Each column in the heatmap represents a population, such that if a gene is hit in that population the square will be colored (darker color if a gene is hit two or more times in that population). Red squares indicate premature-stop-lost mutations in ADE2, which correspond to the populations with asterisks in *Figure 2*. One population that was not sequenced (not shown here) also has this mutation (confirmed by Sanger sequencing). The table at left gives more information on each multi-hit gene: 'High impact' is the fraction of hits that are likely to cause a loss-of-function, as annotated by SnpEff (e.g. nonsense mutations), 'LOH' (loss of heterozygosity) is the fraction of hits in diploid populations that fix homozygously, and 'Effect' describes whether the hits are distributed significantly unevenly across strain-types (S), environments (E), or both (SxE), when compared to a null model where fixations are not strain or environment dependent.

The online version of this article includes the following figure supplement(s) for figure 6:

**Figure supplement 1.** Same as *Figure 6*, but for all multi-hit genes not shown in *Figure 6* (plot 1/3).
**Figure supplement 2.** Same as *Figure 6*, but for all multi-hit genes not shown in *Figure 6* (plot 2/3).
**Figure supplement 3.** Same as *Figure 6*, but for all multi-hit genes not shown in *Figure 6* (plot 3/3).
**Figure supplement 4.** Same as *Figure 6*, but for all multi-hit genes where hits are distributed significantly unevenly across strain-types (S), environments (E), or both (SxE) compared to a null model where fixations are not strain or environment dependent.

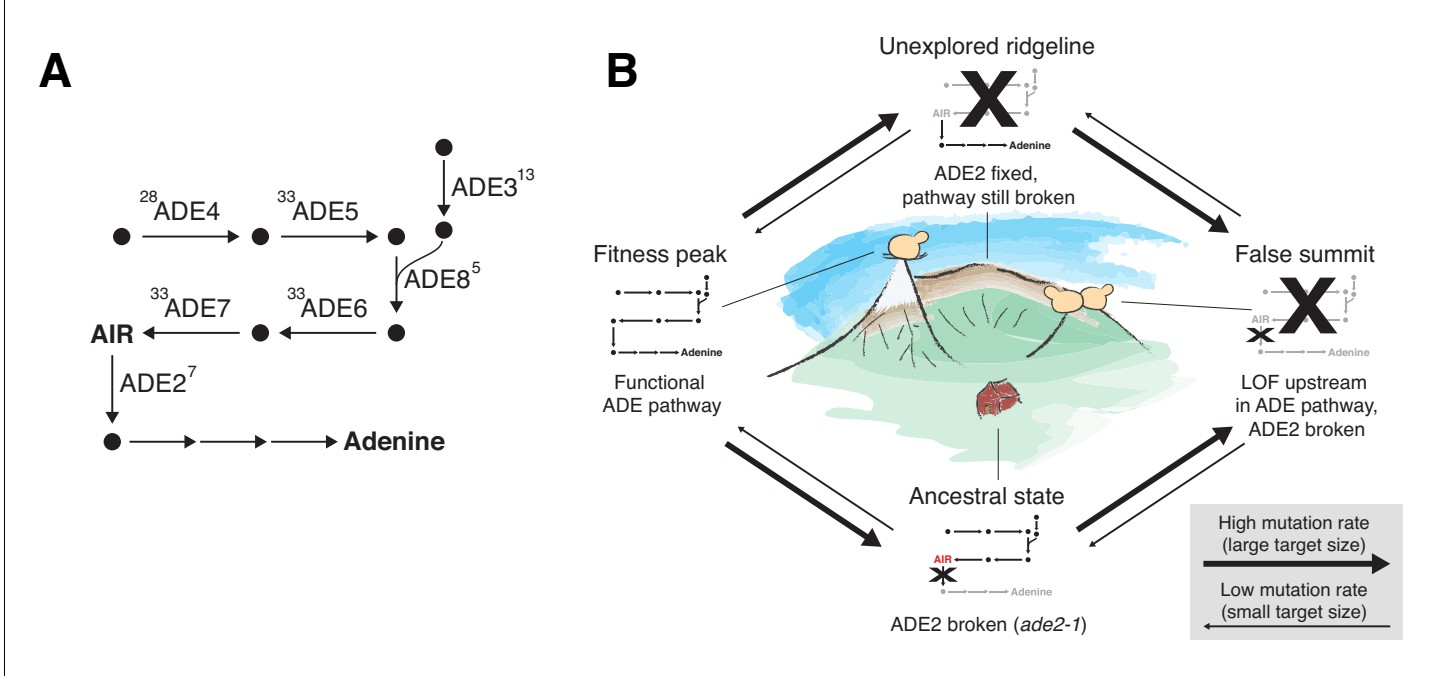

**Figure 7.** ADE pathway evolution. (**A**) Simplified schematic of the adenine biosynthesis pathway. Circles represent metabolic intermediates; AIR is the toxic metabolic intermediate phosphoribosylaminoimidazole. Annotations represent the number of fixed nonsynonymous mutations in each gene (note that ADE5 and ADE7 are both products of the same gene). (**B**) Schematic of a fitness landscape with four possible states defined by whether ADE2 is functional and whether the ADE pathway upstream of ADE2 is functional. The small insets represent the state of the pathway in (**A**) at each position. Elevation in the landscape represents putative fitness differences, and the width of the arrows represents the putative mutation rates between the different states.

The online version of this article includes the following figure supplement(s) for figure 7:

**Figure supplement 1.** Overdispersion.

**Figure supplement 2.** Mutual information analysis.

functional because they disrupt adenine biosynthesis, are strongly beneficial in the *ade2-1* background because they prevent this toxic buildup (*Rojas Echenique et al., 2019*). Consistent with this, we see rapid fixation of at least one mutation in the ADE pathway, typically upstream of ADE2, in almost all of our sequenced populations, along with frequent loss of heterozygosity of these mutations in diploids (*Figure 3A*).

Five of our sequenced populations find a better solution: they fix mutations that revert the premature stop codon so that the full ADE2 sequence can be translated (populations indicated by asterisks in *Figure 2* and mutations shown in red in *Figure 6*; note that one unsequenced high-fitness population also has this mutation, confirmed by Sanger sequencing). These populations have higher fitness than other populations from the same strain background and environment, presumably because they have both repaired the defect in adenine biosynthesis and avoided the buildup of the toxic intermediate. As we would expect, these populations do not fix any loss-of-function mutations in other ADE pathway genes. The fact that only six of our populations find this higher fitness reversion of *ade2-1* is presumably a consequence of differences in target size: while loss-of-function in genes upstream in the pathway can arise from a variety of mutations in five genes upstream of ADE2, the *ade2-1* reversion requires a mutation at a specific codon in ADE2.

We note that once a population has fixed an upstream loss-of-function mutation, it requires reversion of both the original *ade2-1* mutation and the upstream mutation to find the higher fitness genotype. While this is possible in principle, both mutations have single-codon target sizes and when they occur alone are likely neutral and deleterious respectively, making this evolutionary path extremely improbable. We do not observe any populations that move from the lower fitness genotype to the higher fitness genotype even after 10,000 generations of evolution. *Figure 7* depicts these evolutionary states using a simple fitness landscape framework.

## Contingency

The alternative evolutionary paths involving mutations in the ADE pathway are an example of contingency that is already well understood (*Rojas Echenique et al., 2019*; *Roman, 1956*). We next sought to analyze the role of contingency more broadly in our experiment. To do so, we first analyzed whether mutations are over-dispersed or under-dispersed among populations, following *Good et al., 2017*. Looking within each environment-strain combination, we find that nonsynonymous mutations are more over-dispersed than expected by chance; this is still true if we also include mutations that are present but not fixed (*Figure 7—figure supplement 1*). This provides evidence of 'coupon collecting': populations with a fixed nonsynonymous mutation in a gene are less likely to fix another mutation in that gene.

We next sought to test whether mutations in a given gene tend to open up or close off opportunities for beneficial mutations in other genes. To do so, we calculated the mutual information between multi-hit genes (i.e. for each pair of multi-hit genes, whether a population with a fixed nonsynonymous mutation in the first gene is more or less likely to have a fixed nonsynonymous mutation in the second). As in *Fisher et al., 2019*, we find that the sum of mutual information across all pairs of multi-hit genes in our experiment is higher than in simulations (p=0.036, *Figure 7—figure supplement 2*). Thus, there is an overall statistical signature of contingency in our data: mutations in certain genes make mutations in others more or less likely. However, we do not have power to isolate this signature to individual pairs of genes; the mutual information between any two multi-hit genes in our experiment is not higher than we would expect by chance. Note that because we calculate mutual information separately for each environment-strain combination (at most 12 populations per group), we have less power than *Fisher et al., 2019* to detect interactions between genes. In sum, while we cannot confidently identify more specific examples of contingency in our data beyond a general pattern of coupon-collecting, it is likely to be playing a role, as in the LTEE (*Good et al., 2017*).

## Patterns of molecular evolution specific to diploids

Our experiment provides an opportunity to compare asexual adaptation in diploids to that in haploids, and to characterize diploid-specific aspects of the evolutionary dynamics (*Figure 8*). Only one of our focal haploid populations underwent a whole genome duplication and became diploid during our experiment (*Figure 8—figure supplements 1–2*, and see Materials and methods). We were surprised by this result; previous studies have demonstrated that evolving haploid yeast populations often become diploid (*Gerstein et al., 2006*; *Fisher et al., 2018*; *Harari et al., 2018*). We have excluded this population from all comparative analyses, and we restrict our analysis of diploids here to the diploid populations started from a diploid ancestor.

A key difference between evolution in diploids and haploids has to do with the dominance effects of mutations. Some mutations that provide a fitness advantage in haploids may be fully or partially dominant in diploids, and hence provide a fitness advantage when they initially arise in a single chromosome. Others are likely to be recessive, and hence are neutral when they initially arise in a single chromosome. Earlier laboratory evolution experiments have found that diploid populations of budding yeast tend to adapt more slowly than haploids, which could be a signature of the impact of Haldane's sieve (e.g. if most beneficial mutations in haploids are loss-of-function mutations and most loss-of-function mutations are recessive) (*Fisher et al., 2018*; *Marad et al., 2018*; *Zeyl et al., 2003*; *Gerstein et al., 2011*). Consistent with this expectation, our diploid populations did increase in fitness more slowly than haploids over the course of the experiment, and the majority of mutations in our diploid populations fix as heterozygotes. However, we note that diploid populations do not accumulate mutations at a lower rate (and do not have a lower dN/dS) than our haploid *MATα* populations. Thus, the slower rate of fitness increase in diploids could also partly be a consequence of diminishing returns (*Chou et al., 2011*; *Khan et al., 2011*; *Kryazhimskiy et al., 2014*).

One way for recessive (or incompletely dominant) mutations arising in diploid populations to bypass Haldane's sieve is by loss of heterozygosity (LOH), in which a mutation is copied to the sister chromosome by mitotic recombination or whole-chromosome homozygosis (*Forche et al., 2011*; *Gerstein et al., 2014*; *St Charles et al., 2012*). We see signatures of these LOH events across the genome in our experiment (*Figure 8A*). As in *Marad et al., 2018* and *Fisher et al., 2018*, we observe certain areas of the genome with higher rates of LOH, such as the right arms of

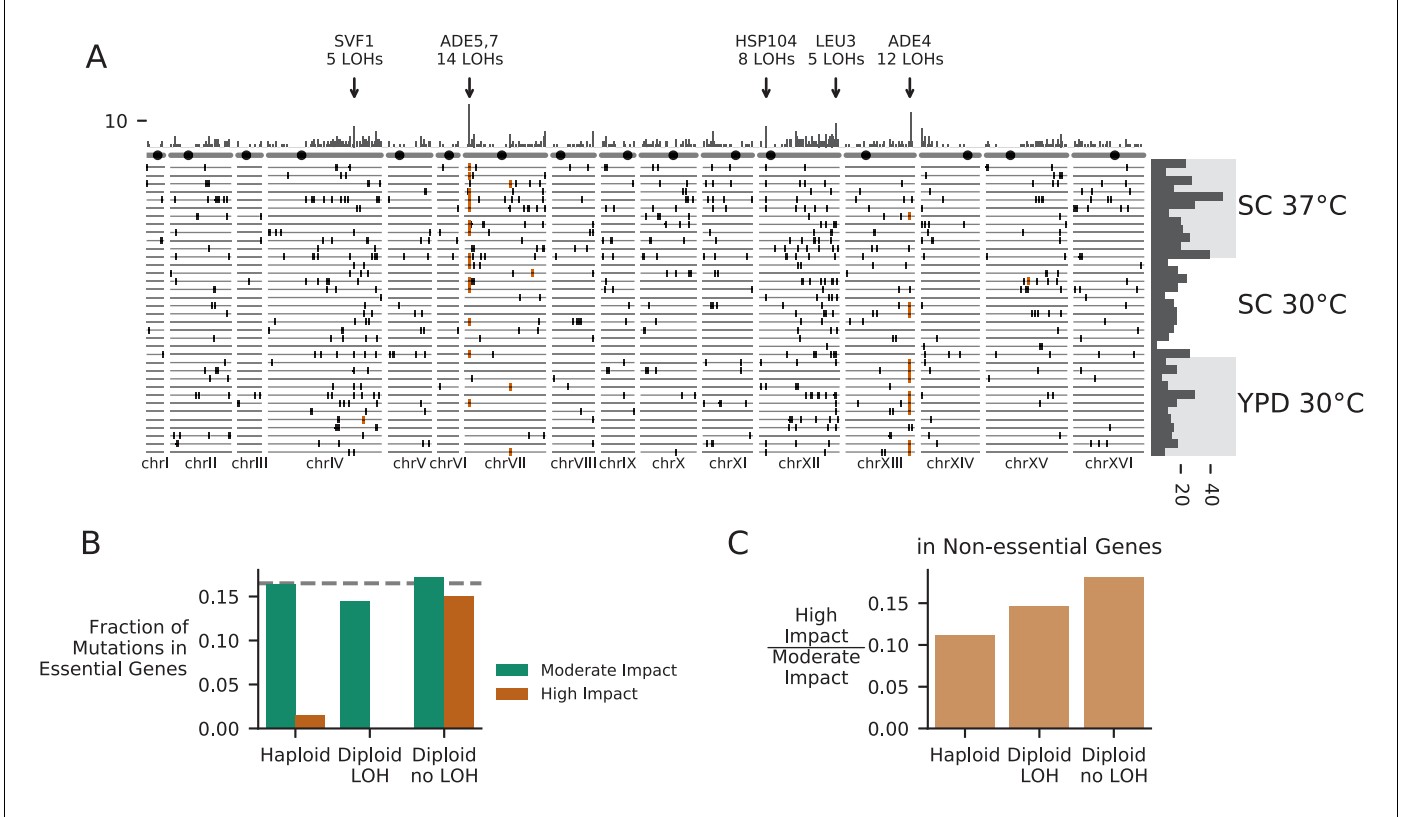

**Figure 8.** Patterns of molecular evolution and loss of heterozygosity in diploids. (**A**) Genomic positions of all mutations that experienced loss of heterozygosity (LOH) across all diploid populations (loss of heterozygosity defined by a mutation reaching >90% frequency). Orange marks represent mutations in the ADE pathway. Each horizontal line represents one population, and the histogram at right represents the total number of LOH fixations in each population, with populations arranged by environment. The top histogram represents the frequency of loss of heterozygosity across the genome, and the chromosomes underneath show the centromere location with a black circle. Genes with five or more LOH fixations are annotated. (**B**) The fraction of fixed nonsynonymous mutations that are in essential genes, plotted for mutations fixed in haploid populations, mutations fixed homozygously in diploid populations (LOH) and mutations fixed heterozygously in diploid populations, plotted separately for mutations annotated as high or moderate impact by SnpEff (high-impact mutations are likely to cause loss-of-function). The dashed line represents the fraction of the coding genome that is in essential genes. (**C**) The ratio of high-impact to moderate-impact fixations in the same three mutation groups as in (**B**), for mutations in non-essential genes only.

The online version of this article includes the following figure supplement(s) for figure 8:

**Figure supplement 1.** The ploidy state of two clones from each focal population, shown by FITC histograms of Sytox-stained cells.

**Figure supplement 2.** Cell imaging from three populations with abnormal Sytox data.

chromosomes XII and IV. These concentrations of LOH are likely due to some combination of selection in favor of LOH events and differences in the rates at which they occur. Higher rates of LOH on the right arm of chromosome XII are likely related to high levels of recombination associated with the ribosomal DNA array (*Fisher et al., 2018*; *Marad et al., 2018*), but we also see evidence that patterns of LOH are affected by selection for recessive beneficial mutations that would otherwise be filtered out by Haldane's sieve (as in *Gerstein et al., 2014*), notably among loss-of-function mutations in the adenine pathway (*Figure 3A*, *Figure 8A*).

As driver mutations sweep to fixation in diploids, they have the potential to bring along recessive deleterious hitchhikers (which then also fix as heterozygotes). Consistent with this, we find that mutations fixed as heterozygotes in diploids include a large percentage of high-impact mutations in essential genes, while mutations fixed in haploids and mutations fixed homozygously in diploids include nearly zero high impact mutations in essential genes (*Figure 8B*). Even in non-essential genes, mutations that fix heterozygously in diploids are more likely to be high-impact mutations compared to those that fix in haploids or those that fix homozygously in diploids, again suggesting that diploids are fixing recessive deleterious mutations as heterozygotes (*Figure 8C*). This build-up

of recessive deleterious load in diploids is expected, but takes on an interesting light in the context of the widespread loss of heterozygosity we observe. As recessive deleterious load accumulates in the population, it will limit the rate of LOH by making many LOH events strongly deleterious or lethal. Thus, passage through Haldane's sieve by loss of heterozygosity should become less likely as populations accumulate a substantial load of hitchhiking heterozygous mutations. This process is likely to be occurring in domesticated industrial diploid yeast lineages, which often become obligately asexual and accumulate many heterozygous mutations (*Gallone et al., 2016*). However, we note that in sexual lineages recombination with sufficient inbreeding could dramatically alter these dynamics, by continuously purging recessive deleterious load (*Charlesworth and Willis, 2009*).

While we hypothesize that most of the heterozygous fixations in diploids are either dominant beneficial mutations or neutral or deleterious hitchhikers, some may be overdominant beneficial mutations. One possible candidate for overdominance is CCW12, which is hit preferentially in diploids (*Figure 6*) and in which only 2 of the 17 fixed mutations lost heterozygosity (both these mutations are in-frame deletions of the final amino acid, and note that CCW12 is in a region on the right arm of chromosome XII where LOH appears common). In *Leu et al., 2020*, mutations in CCW12 were maintained in asexual diploid populations but lost in sexual populations, supporting a hypothesis of overdominance, although *Leu et al., 2020* did not detect overdominance in reconstructed strains in their evolution environments. Extensive reconstructions or backcrossing will be required to understand the importance of overdominance in the evolution of our diploid populations.

## Loss of the 2-micron plasmid and killer phenotype

While the mitochondria is maintained throughout evolution in all of our focal populations, we detected frequent loss or loss-of-function in three other exclusively extrachromosomal elements: the 2-micron plasmid and two double-stranded RNA components of the yeast 'killer virus' toxin-antitoxin system (*Schmitt and Breinig, 2002*).

Following *Buskirk et al., 2020* and *Jerison et al., 2020*, we performed halo killing assays against a sensitive strain for each of our focal populations at each sequenced timepoint (*Woods and Bevan, 1968*). To avoid potential inactivation of the toxin at higher temperatures in liquid media (*Woods and Bevan, 1968*), we conducted these assays at room temperature on agar plates. By the final timepoint, 89/90 of our focal populations lost the ability to kill a susceptible strain (*Figure 9A*), and this loss happened most rapidly in the high temperature environment, consistent with previous work (*Jerison et al., 2020*). This effect is likely due to some combination of segregation or replication failure at high temperatures (*Weinstein et al., 1993*) and toxin inactivation at high temperatures (*Woods and Bevan, 1968*). While we have not sequenced RNA viral genomes over time in our experiment, *Buskirk et al., 2020* also observed widespread loss of killing ability in evolved yeast populations and determined that mutations in the K1 toxin gene were causing a loss of killing ability. While they found no evidence for a fitness benefit to the host from these loss-of-function mutations in the toxin gene, they observed that these mutations were favored in intracellular competition with other viral variants.

The 2-micron plasmid is a selfish genetic element that imposes a cost on the cell without providing any apparent benefit (*Harrison et al., 2012*). Because it is a DNA element, we can directly observe loss of this element in many of our populations. It appears to be lost less frequently in diploid strains, and, as with the killer phenotype, it is lost most consistently and rapidly in the high-temperature environment (*Figure 9B*).

## Discussion

Evolution experiments are as much about hypothesis generation as hypothesis testing, and work across the field has now laid out a series of hypotheses about evolution in general. No experiment can cover the breadth of biological and environmental diversity needed to fully test these hypotheses; we cannot replay all of evolution. However, a relatively consistent set of results has emerged across microbial species evolved asexually for thousands of generations in the lab (*Kassen, 2014*). Our results confirm many aspects of the picture drawn by previous work, with several important exceptions.

Most of our populations followed predictable fitness trajectories in which fitness increases slowed over time. This pattern was not observed, however, in some of our diploid populations in SC 30°C,

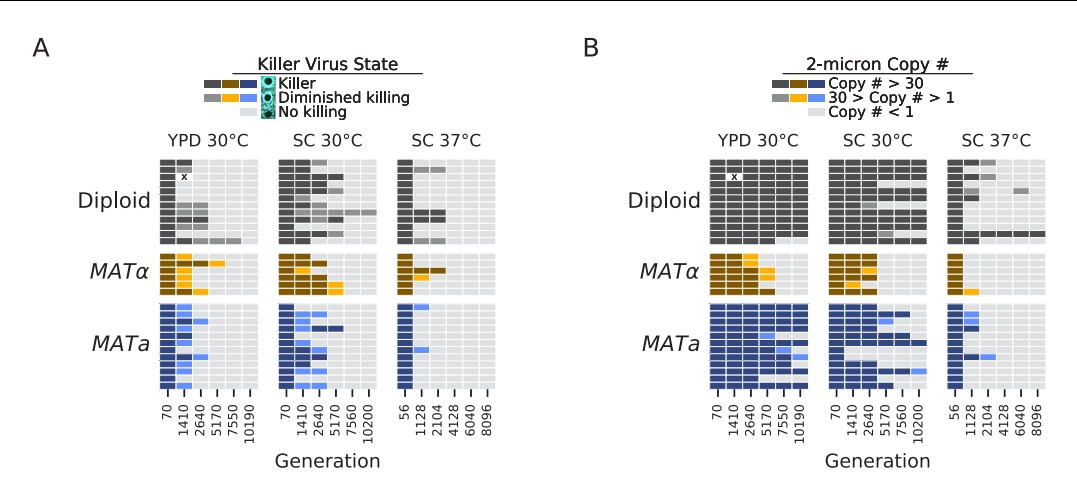

**Figure 9.** Loss of extrachromosomal elements. (**A**) Killer virus activity at each sequenced timepoint, determined by a killer assay against a sensitive strain. Each row represents one population. Examples of raw data for each qualitative phenotypic category are shown in the key, and the full raw data underlying these scores is shown in *Figure 9—figure supplement 1*. (**B**) 2-micron plasmid copy number at each sequenced timepoint. Rows represent the same populations as in **A**. The x in a diploid population at generation 1410 marks a population we excluded due to contamination in the population during these experiments.

The online version of this article includes the following figure supplement(s) for figure 9:

**Figure supplement 1.** Contrast-enhanced scanned images of killer virus halo assays.

which instead increased in fitness at a slow constant rate similar to *Marad et al., 2018*, before experiencing significant rapid increases in fitness likely associated with individual selective sweeps (*Figure 2*). Our populations show signatures of clonal interference, and they accumulated fixed mutations linearly through time even late in the experiment. We find only one strong case of repeatable selection at the level of the nucleotide change (*ade2-1* reversions), but we observe widespread parallelism across strains and environments at the level of genes and pathways: populations predictably adapt through loss-of-function mutations in the adenine biosynthesis pathway, sterility-associated genes, and negative regulators of the Ras pathway.

We do not observe two phenomena that results from the LTEE had previously suggested might be common: the fixation of mutator alleles that dramatically increase mutation rates, and the spontaneous emergence of long-term quasi-stable coexistence between competing lineages. The reasons for these differences remain unclear. In part, we may not observe these phenomena simply because of the shorter timescale of our experiment. However, we note that within the first 10,000 generations of the LTEE, 4 of the 12 populations fix mutator alleles, and 3 of the 12 populations have coexisting lineages detectable from sequencing data. Instead, the lack of mutator lineages may stem from a difference in the rate at which mutators arise or a different balance between the relative importance of beneficial and deleterious mutations (which depends on the environment and ancestral fitness; see e.g. *Swings et al., 2017* and *Kryazhimskiy et al., 2014*) that leads to less indirect selection for mutators (*Good and Desai, 2016*). We also may have less second-order selection for mutators in our experiment because our strains have mutation rates that are higher than the ancestral *E. coli* strain in the LTEE, though lower than the LTEE mutator lineages (*Lang and Murray, 2008*; *Wielgoss et al., 2013*). The difference in how commonly coexistence emerges is similarly unclear. Our strains and environments may simply lack the metabolic pathway architecture to produce cross-feeding or other interactions that could be the basis for coexistence. Alternately, coexisting lineages may have emerged in our experiment but been lost due to drift or strong within-lineage adaptation. Since our populations are smaller than those in the LTEE, low-frequency lineages will be lost more commonly during the daily bottleneck. Regardless of the reasons for these differences, our results suggest that the evolution of mutation rates and of stable ecological interactions may not be as general or widespread as the LTEE has suggested, and may instead vary substantially based on differences in the organisms or details of the environmental conditions.

As the longest running evolution experiment in yeast, this project provides a window into how dominance and loss of heterozygosity can affect the dynamics of adaptation in diploids. Our diploid populations appear to carry substantial recessive deleterious load (*Figure 8B–C*) and may carry beneficial overdominant mutations, but future studies involving genetic reconstructions or backcrossing will be needed to fully characterize these effects. We also observe widespread loss of heterozygosity. The dynamics of mutations in the adenine biosynthesis pathway provides a particularly interesting example of both how Haldane's sieve slows adaptation in diploids and how diploids can bypass the sieve by loss of heterozygosity. At some point during the experiment, most of our diploid populations homozygously fix a loss-of-function mutation upstream in the pathway, which eliminates the deleterious toxic intermediate produced as a result of the ancestral *ade2-1* mutation. While the rate of loss of heterozygosity was high enough to produce these genotypes and expose them to selection, it appears to have been a limiting factor; haploids typically fixed these mutations earlier in the experiment (*Figure 3—figure supplements 1–9*). Haldane's 'speedcheck' here slowed adaptation but also provided diploid populations with more time to search for the single-codon target of the (highly beneficial and apparently dominant) *ade2-1* reversion, and indeed, 4/6 populations with this mutation in our experiment are diploids.

Perhaps, the most important product of microbial evolution experiments is a base of intuition for understanding how the interactions between different evolutionary forces determine the dynamics and outcomes of genotypic and phenotypic evolution. The extent to which this base of intuition can be generalized across systems and scales – ranging from specific protein complexes to human pathogens to entire clades of sexually reproducing species – is an important set of largely unanswered questions. However, laboratory microbial evolution experiments have provided basic expectations to compare against, and have highlighted a collection of phenomena that can sometimes play a major role in adaptation. Our results here reinforce the conclusion that long-term adaptation to a constant environment can be characterized by widespread clonal interference, contingency, and steady molecular evolution even as fitness increases slow down over time. They also highlight the role of dominance and loss of heterozygosity in diploid evolution. However, our work also calls into question the generality of conclusions about the importance of the evolution of mutation rates or stable coexistence. As our populations continue to evolve, further analysis of our experiment and of other complementary studies will further broaden our understanding of the processes that determine the rate, predictability, and molecular basis of evolution.

## Materials and methods

### Strains

The two haploid strains used for this study are MJM361, which has genotype *MATa*, YCR043C: KanMX, STE5pr-URA3, *ade2-1*, his3Δ::3xHA, leu2Δ::3xHA, trp1-1, can1::STE2pr-HIS3 STE3pr-LEU2, HML::NATMX, *rad5-535* and MJM335, which has genotype *MATα*, YCR043C:HGHB, STE5pr-URA3, *ade2-1*, his3Δ::3xHA, leu2Δ::3xHA, trp1-1, can1::STE2pr-HIS3 STE3pr-LEU2, HMR::NATMX, *rad5-535*. We created MJM361 and MJM335 by knocking out HML or HMR with the NatMX cassette in MJM64 and MJM36 respectively (*McDonald et al., 2016*). The diploid strain used in this study, MJM102, is a cross of MJM64 and MJM36.

### Culture conditions

We propagated all populations in 128 μL of media in unshaken flat-bottom polypropylene 96-well plates (VWR #82050–786). For one environment, we used rich YPD media (1% Bacto yeast extract (VWR #90000–726), 2% Bacto peptone (VWR #90000–368), 2% dextrose (VWR #90000–904)) and grew populations at 30°C. For the other two environments, we used synthetic complete (SC) media (0.671% YNB with nitrogen (Sunrise Science #1501–250), 0.2% SC (Sunrise Science # 1300–030), 2% dextrose) and grew populations at 30°C or 37°C. All media was supplemented with 100 μg/ml ampicillin and 25 μg/ml tetracycline. Using a Biomek FXp robot (Beckman Coulter), we performed daily 1:2^10 dilutions of populations in YPD 30°C and SC 30°C and daily 1:2^8 dilutions of populations in SC 37°C (we used 384-well plates (VWR #82051–306) for serial dilution). These dilutions determine the number of doublings or generations per day (10 for YPD 30°C and SC 30°C, eight for SC 37°C), the bottleneck population size (~8 • $10^3$ for YPD 30°C and SC 37°C, ~2 • $10^3$ for SC 30°C), and the

corresponding effective population size (~6 • $10^4$ for YPD 30°C, ~4 • $10^4$ for SC 37°C, ~1 • $10^4$ for SC 30°C) (**Wahl and Gerrish, 2001**). These bottleneck sizes are based on estimated saturation densities of ~6 • $10^7$ cells/mL for YPD and ~1.6 • $10^7$ cells/mL for SC in 96-well plates, measured using a Coulter Counter Z2 (Beckman Coulter). Before dilution, we resuspended cultures by shaking at 1200 rpm for 2 min, and after dilution we shook the new plates at 1200 rpm for 1 min, both on a Titramax 100 plate shaker (Heidolph Instruments). After each transfer, the tips (VWR #89204–794) used to dilute cultures were washed with water (to wash out cells) and 100% ethanol (to lyse residual cells), left to dry overnight, and reused in culture propagation. The 96-well microplates used to maintain populations were bleached (to lyse cells), washed with distilled water, and autoclaved (121°C, 30 min) before being reused. Every 7 days, we froze aliquots of all populations in 27% glycerol (final concentration) at −80°C. To monitor for contamination, six well-spaced wells in each environment were intentionally left 'blank' at the start of the experiment (i.e. they contained only media and no cells). At several timepoints during the evolution we noticed contamination in the previously blank wells of our 96-well plates. During instances of contamination, we unfroze all populations from an older glycerol archive and inoculated 4 μL directly into 124 μL of the appropriate media for each environment. A record with notes on the evolution is available in **Supplementary file 1**.

## Population loss and cross-contamination

Over the course of this experiment, we periodically screened our populations for drug resistance (Hygromycin, G-418, and ClonNat) in order to detect cross-contamination. Using these checks, we observed multiple instances of cross-contamination in the YPD 30°C environment from other yeast species which were part of a concurrent evolution experiment. These events were likely due to mistakes during our tip washing or media filling procedures, which were more carefully controlled later in the experiment. As described above for the case of outside contamination, if we observed cross-contamination shortly after it occurred, we restarted the experiment from a previously frozen timepoint. In several cases, we failed to recognize cross-contamination until thousands of generations had passed, so we excluded these populations from our analysis (these populations are listed in **Supplementary file 1**). While it is possible that cross-contamination occurred more frequently than just the cases we observed, the sequencing data for focal populations suggests otherwise (fixed mutations remained fixed in all populations for the duration of the experiment). Due to a combination of errors in Biomek pipetting and evaporation (especially in the 37°C environment), we also lost several populations over the course of the experiment (the wells became blanks); these are also excluded from our analysis, and are listed in **Supplementary file 1**.

## Fitness assays

In order to assess competitive fitness using a consistent reference for each environment, we isolated clones at various generations from an arbitrarily chosen evolving diploid population in YPD 30°C (P1G09). We looked for clones that had fitnesses intermediate between the ancestral strains and evolved strains in each environment, and tagged these clones by inserting a yNatMX cassette and GFP (pRPL39::eGFP::tADH) into an intergenic region (chromosome VII, position 649234) that was previously used as a neutral insertion site control in **Johnson et al., 2019**. This produced the reference strains used for fitness assays in YPD 30°C and SC 30°C (2490A-GFP1), and SC 37°C (11470A-GFP1).

Fitness assays were performed as described previously (**Lang et al., 2011**). Briefly, we unfroze populations and a reference strain from glycerol stocks, allowed them to grow in their evolution environment for one full growth cycle, and then mixed the populations with the reference strain in equal proportions. We then maintained these mixed populations for three daily growth cycles, as described above. At each transfer, we diluted cells from each well into PBS and used flow cytometry (Fortessa and LSRII, BD Biosciences) to measure the ratio of the two competing types, counting approximately 10,000–40,000 cells for each measurement.

To get fitness measurements for each population-timepoint, we first calculated the frequency of fluorescent reference cells in each sample by gating our flow cytometry data to separate the fluorescent cells. Because a small percentage of reference cells do not fluoresce, we estimated this percentage from six wells that only contained the reference in each environment and used these values to correct the reference frequency in all other wells. We then calculated the fitness of each population-

timepoint as the slope of the natural log of the ratio between the frequencies of the non-reference and reference cell populations over time (timepoints with reference frequency under 5% or over 95% were excluded). After taking the mean of fitness measurements from two replicates, we corrected for batch effects in our assays by subtracting the mean fitness measured for an unlabeled reference (2490A) in the same fitness assay. While our replicate fitness measurements generally correlate very well (*Figure 2—figure supplement 2*), there is a deviation from the 1:1 line for replicate fitness measurements for low-fitness populations in SC 37°C. We believe this is due to batch effects specific to each flow cytometry machine, which in turn affect the reference frequency correction explained above. Only assays in which the reference frequency goes up very rapidly (low-fitness populations) are strongly affected.

Because the ancestral genotypes have a strongly deleterious mutation in the adenine biosynthesis pathway and haploids very quickly fix strongly beneficial suppressor mutations, it was difficult to measure ancestral fitness in some cases; we sometimes observed changes in fitness during the fitness assay even when using clones from generation zero (which had been grown prior to and after freezing glycerol stocks). In all but one environment-strain combination, we were able to identify populations without any nonsynonymous mutations detected from our sequencing data at the first timepoint (generation 70 for YPD 30°C and SC 30°C, generation 56 for SC 37°C), so we used the median of the fitness measured among these populations at the first timepoint to define ancestral fitness. All *MATa* populations in YPD 30°C had nonsynonymous mutations present (and often fixed) at generation 70, but one unsequenced population had a significantly lower generation 70 fitness than all others (similar to the one *MATα* population in YPD 30°C with no-nonsynonymous mutations at generation 70), so we use the fitness estimated at generation 70 for that population as our ancestral fitness for *MATa* populations in YPD.

## Whole-genome sequencing

For each of the three environments, we selected 30 focal populations: 12 diploid populations, 12 *MATa* populations, and 6 *MATα* populations. We chose these populations randomly after excluding populations in wells along the edge of the plate (which we have had the most problems with losing due to evaporation or pipetting errors) and populations where we had detected cross-contamination. We performed whole-genome, whole-population sequencing on each of these populations at six timepoints. After unfreezing populations as described above, we transferred each of our focal populations into five replicate wells in their evolution environment, let them grow for 24 hr, and then pelleted ~0.5 mL of cells. We used a DNA extraction protocol based on the 'BOMB gDNA extraction using GITC lysis' from *Oberacker et al., 2019*. Briefly, we resuspended the cell pellets in 50 µL of zymolyase buffer (5 mg/mL Zymolyase 20T (Nacalai Tesque), 1M Sorbitol, 100 mM Sodium Phosphate pH 7.4, 10 mM EDTA, 0.5% 3-(N,N-Dimethylmyristylammonio)-propanesulfonate (Sigma, T7763), 200 µg/mL RNAse A, and 20 mM DTT) (*Nguyen Ba et al., 2019*) and incubated the suspension at 37°C for 1 hr. Subsequently, we added 85 µL of a modified BOMB buffer (4M guanidinium-isothiocyanate (Goldbio G-210–500), 50 mM Tris-HCl pH 8, 20 mM EDTA) and then 115 µL of isopropanol (VWR# BDH1133-4LP), mixing by pipetting for 3 min after each addition. We then added 20 µL of Zymo Research MagBinding beads to bind DNA, mixed for 3 min by pipetting, separated beads from the solution using a Magnum FLX 96-well magnetic separation rack (Alpaqua), and removed the supernatant. We washed the beads with 400 µL of isopropanol and twice with 300 µL of 80% ethanol. Finally, we added 75 µL of sterile water to the beads and mixed by pipetting for 3 min. Finally, we separated beads from solution and transferred 44 µL of the supernatant (containing the DNA) into a new 96-well PCR plate (Bio-Rad HSP9631) for library preparation. This entire process was carried out on a Biomek FXp robot (Beckman Coulter).

Sequencing libraries were prepared using a Nextera (Illumina) kit as previously described (*Baym et al., 2015*), but with three additional PCR cycles for a total of 16, and with a two-sided bead-based size selection after PCR (we used either 0.5/0.7X or 0.55/0.65X bead buffer ratios with PCRClean DX Magnetic Beads (Aline)). Libraries were sequenced to an average depth of 20-fold (haploids) or 40-fold (diploids) coverage using a NextSeq 500 or Novaseq (Illumina).

## Sequencing analysis

We trimmed Illumina reads with NGmerge version 0.2 (*Gaspar, 2018*), aligned all the first-timepoint samples to a SNP-corrected W303 genome (*Lang et al., 2013*) using BWA version 0.7.15 (*Li, 2013*), and marked duplicate reads with Picard version 2.9.0 (http://broadinstitute.github.io/picard). We used samtools (*Li et al., 2009*) to merge these alignments and then used Pilon version 1.23 (*Walker et al., 2014*) to create a new reference genome that is corrected for additional SNPs present in the ancestral strains. We then repeated this process until the marking duplicate reads step for all samples using this new reference and called variants using GATK version 4.1.3.0 (*McKenna et al., 2010*), specifically using HaplotypeCaller, GenomicsDBImport, and GenotypeGVCFs with heterozygosity set to 0.005. We annotated these variants using SnpEff version 4.3T (*Cingolani et al., 2012*), and split multi-allelic records into individual records.

We extracted allele depths for each variant to determine the number of reads supporting the reference and alternate alleles at each site. We then filtered variants based on these read counts. We first excluded mutations with less than five reads representing the alternate allele across all timepoints. To create this filtered list of variants present in each population, we required that mutations pass at least one of these two criteria:

1. The total alternate-allele reads across all timepoints for the population in question is more than 90% of the total alternate-allele reads across all populations and *all* timepoints.
   OR
2. At least two timepoints have at least five reads supporting the alternate allele AND The total alternate-allele reads across all timepoints for the population in question is more than 90% of the total alternate-allele reads across all populations at *only the first timepoint.*

The first criterion addresses if a mutation is unique to a single population, which provides strong evidence that it is not a common sequencing or alignment error. However, we do not want to exclude the possibility of parallelism at the nucleotide level, so we include the second criterion as a more lenient way to exclude these types of errors while not requiring uniqueness. Some small number of sequencing or alignment errors will pass these filters, so we emphasize that this is only a lenient first step, and that our analysis of parallelism and contingency relies on also observing fixation.

We simplify our SnpEff annotations to indicate one of five types of mutation; in order of decreasing putative effect they are indel, nonsense, missense, synonymous, or noncoding. For mutations with multiple annotations, we assign the mutation type with the largest putative effect. To test if some nearby mutations are part of a single mutational event, we perform Fisher's exact test on the alternate and reference allele counts at each timepoint for mutations within 25 bp of each other. If two mutations have no significant differences detected at the $p < 0.01$ level for any timepoint, we label them as part of the same 'mutation group,' and they are counted as one mutation in subsequent analysis. We define mutations as 'present' at a particular timepoint if they have coverage of at least 5X and are at greater than or equal to 0.1 frequency. We define mutations as 'fixed' at a particular timepoint if they have coverage of at least 5X, are at greater than or equal to a frequency of 40% (diploids) or 90% (haploids), and do not drop below these thresholds while still at >= 5X coverage at a later timepoint. If a mutation is called fixed at one timepoint, it is automatically called fixed at later timepoints, even if they have less than 5X coverage. Using the same rules, we also call loss of heterozygosity of a mutation in diploids using a frequency threshold of 90%. We exclude mutations called in the 2-micron plasmid from further analysis since most populations lose this plasmid during evolution, and variation in coverage and misalignments can easily produce false mutation calls in these cases. We also exclude mutations in the telomeres, where alignment errors and repetitive regions make mutation calling difficult.

## Structural variant / copy number variant analysis

We use LUMPY and smoove (*Pederson, 2020*) to call structural variants in our sequencing data (*Layer et al., 2014*). All structural variants called are listed in the processed variant call files for each population included in *Supplementary file 2*. In addition, we use a custom pipeline to identify putative copy number variants (CNVs) in our data. We use samtools-depth to calculate per-site depth from our bam files, and then calculate the average depth in non-overlapping 500 bp windows along the genome. We calculate the median window-depth across the entire genome for each sample and

divide all window-depths by this value to get 'relative depth.' To account for regions that are at a different copy number in our ancestral strains, we calculate the average relative depth at the first sequenced timepoint for each window (and for each strain). We divide the relative depth in our data by these values to get 'standardized depth.' Windows with a relative depth less than 0.25 at the first timepoint are excluded from analysis. For each chromosome in each sample, we use a simple, untrained HMM to detect tracts of standardized depth that deviate from the expectation of 1. We allow states 0, 0.5, 1, 1.5, 2, 3, and 4, with variances equal to the calculated variance in standardized depth multiplied by the state (except for state 0, where we use the calculated variance multiplied by 0.5), initial probabilities of 1% for each non-1 state (94% for state 1), and transition matrix probabilities of 0.01% for all non-diagonal entries (99.94% along the diagonal). This is a rough detection method, but it succeeds in identifying putative CNVs, which we then subject to a filtering process. First, we merge CNV records across timepoints if they cover the same region. Next, we exclude CNVs in telomeric regions, CNVs found in only one timepoint, and CNVs that are less than four windows (2 kb) long. Finally, we manually inspect our structural variant and CNV calls together using a modified version of Samplot (*Belyeu et al., 2020*) to create a list of confirmed copy number variants in our populations (*Supplementary file 3*). During this analysis, we noticed two regions with high copy-number that experienced copy-number changes in many populations: one associated with the *CUP1* tandem array and one associated with the ribosomal DNA tandem array. We excluded these regions from the above analysis and show their copy number changes in every population in *Figure 3—figure supplement 11*.

## Analysis of multi-hit genes

To look for evidence of parallelism and contingency in our data, we focus on nonsynonymous mutations in genes (we consider all open reading frames to be genes for this analysis) that are fixed at the final timepoint. We define the multiplicity as the number of hits multiplied by the number of possible nonsynonymous mutations for that gene, scaled by the mean number of possible nonsynonymous mutations across all genes. In our null model, the hits for each population (the number of hits from our data) are randomly assigned to genes from the complete set of 6579 annotated open reading frames in *S. cerevisiae*, weighted by the number of possible nonsynonymous mutations in each gene. We simulate these random draws 1000 times to generate null distributions for gene multiplicity and number of unique populations in which a gene has at least one hit. To look for parallelism at the codon level, we randomize the location of each nonsynonymous fixed mutation in the gene in which it occurred and then count the number of populations with simulated hits at each amino acid position. We repeat this process ten times to build a null distribution and compare it to the empirical distribution of populations hit for each amino acid position (*Figure 5C*).

Next, we move away from using multiplicity, since we know that selection is playing a large role and mutation rate is not completely limiting adaptive dynamics (clonal interference is observed in our sequencing data). This means that as we look to identify common targets of selection, we will treat the probability of a hit in any gene where we have observed at least one hit as equally likely (instead of weighting by the number of possible nonsynonymous mutations). To this end, we define multi-hit genes as those with hits in at least six populations. Based on the simulations above, a gene has a p=0.02 chance of being hit in at least six populations. While this is a lenient cutoff that will produce a number of false positives, we see a large excess of genes with hits in greater than six populations in our data (*Figure 5B*). To look for functional patterns in our mutation data, we performed GO-Term Enrichment on the set of multi-hit genes analysis using the GOATOOLS python library (*Klopfenstein et al., 2018*). The enrichments that were significant after Benjamini-Hochberg multiple hypothesis testing correction are listed in *Supplementary file 4*.

We define essential genes based on data from the yeast gene deletion collection (*Giaever et al., 2002*; *Liu et al., 2015*). The set of essential genes used in *Figure 8* are those classified as non-evolvable in *Liu et al., 2015*.

## Multi-hit gene enrichment by strain and/or environment

Within a given set of populations, we model the probability of a multi-hit gene *i* being associated with a fixation event based on Ñ$_i$, the number of populations with a mutation fixed in gene *i* plus a 0.1 pseudocount (to avoid zero probabilities), and *M*, the total number of gene hits across all

populations in the set (we ignore when a gene is hit multiple times in the same population for this probability calculation). We model the probability a gene is hit in population $g$ based on the total number of gene hits in population $g$, $M_g$, where $h_{gi}$ is 1 if gene $i$ is hit in population $g$, and 0 if not:

$$P(Gene\ i\ not\ hit\ in\ pop\ g) = P(h_{gi} = 0) = (1 - (\tilde{N}_i / M))^{M_g},$$

$$P(Gene\ i\ hit\ in\ pop\ g) = P(h_{gi} = 1) = 1 - (1 - (\tilde{N}_i / M))^{M_g}.$$

To test whether genes are disproportionately hit in different strain backgrounds (*MATa*, *MATα*, diploid) or different environments, we compare four models that use different sets of populations to compute $P(h_{gi})$:

1. $P(h_{gi})$ is calculated using the entire set of populations (so that there is only one $P(h_{gi})$ for each multi-hit gene)
2. $P(h_{gi})$ is calculated separately for each strain background (so that there are three $P(h_{gi})$ for each multi-hit gene, one for each strain)
3. $P(h_{gi})$ is calculated separately for each environment (so that there are three $P(h_{gi})$ for each multi-hit gene, one for each environment)
4. $P(h_{gi})$ is calculated separately for each environment-strain combination (so that there are nine $P(h_{gi})$ for each multi-hit gene, one for each environment-strain combination)

For each multi-hit gene, we calculate the log-likelihood of the data under each model and calculate log-likelihood ratios between model 1 and each of the other three models. We then create 10,000 null datasets by drawing values from the probabilities defined in model 1 and compute log-likelihood ratios for these simulated data. To define significant effects (at a p<0.05 level), we compare our log-likelihood ratios to distributions of log-likelihood ratios from these null datasets and correct for multiple hypothesis testing using a Benjamini-Hochberg correction. If multiple models are significantly better than model 1, we use the Akaike information criterion (AIC) to determine the model that best explains the data. Data on multi-hit genes and these statistical tests are available in in *Supplementary file 4*.

## Mutual information analyses

Next, we investigate whether the fixation of a mutation in any of our multi-hit genes is dependent on fixation of a mutation in another multi-hit gene. For each environment-strain combination, we calculate mutual information between all multi-hit genes as described in *Fisher et al., 2019*. To avoid mistaking an environment or strain effect for an association between genes, we use separate probabilities for each environment-strain combination, as in model four above, and define the probability of a mutation having a hit in gene $i$ in any one population from environment-strain combination $e$ as (equation 2.1 from *Fisher et al., 2019*):

$$P(h_i | e) = C_M \sum_{g=1}^{N_e} \tilde{M}_{gi}$$

where $\tilde{M}_{gi} = M_{gi} + \varepsilon$ and $M_{gi}$ = 1 if there is a hit in gene $i$ in population g, $N_e$ is the total number of populations in environment-strain combination $e$, $C_M$ = 1/N(1 + $\varepsilon$), and the pseudocount $\varepsilon$ = 1/M (M is the total number of gene hits across all populations in the environment-strain combination, as in model 4 above). In contrast to the enrichment analysis above, this formula treats populations as exchangeable, which is a reasonable assumption since the number of fixed mutations in each environment-strain combination is not highly variable (*Figure 3B*).

We calculate joint probabilities and mutual information as in *Fisher et al., 2019* (see equations 2.2-2.7). Because we separate our data into sets of populations with a shared environment and strain background, our data contain many cases where a gene is hit zero times, which inflates the sensitivity to the pseudocount used in *Fisher et al., 2019*. To avoid this issue, we set the mutual information between two genes to zero if either of the genes has no mutations in a given environment-strain combination. We sum the mutual information values for each pair of genes across the nine possible environment-strain combinations, and record the total mutual information (*MI*<sub>tot</sub>, the sum of *MI*

values across all possible gene pairs) and the maximum mutual information between any two genes ($MI_{max}$).

Next, we compare these results to simulated datasets. We create 10,000 null datasets by drawing from $P(h_i|e)$ for each gene, and calculate mutual information as described above to build null distributions for $MI_{tot}$ and $MI_{max}$.

The results are plotted in *Figure 7—figure supplement 2*. While $MI_{tot}$ for our data is higher than in simulated datasets (p=0.036), $MI_{max}$ for our data lies well within the range of simulated data, so we cannot detect any specific examples of contingency. As in *Fisher et al., 2019*, we test the robustness of our results to choices of the pseudocount $\varepsilon M$ between 0.1 and 2 (the value used above was 1), and find that it does not qualitatively change our results.

## Over-/under-dispersion analysis

Following *Good et al., 2017*, we looked for statistical patterns of contingency by comparing the dispersion configurations for genes with simulated data. For each environment-strain combination, we record the number of times each gene is hit and the number of populations in which it is hit. We also simulate distributing these hits across populations by multinomial draws weighted by the number of hits in each population. We run this simulation, for each possible number of hits (up to the maximum observed), 10,000 times for each environment-strain combination. For each number of hits, we compute the probability of those hits being distributed among each possible number of populations for both our data and the simulated data. We compute the 'excess probability' in our data by subtracting the simulated probability from the data probability. The results are plotted in *Figure 7—figure supplement 1A*. We repeat this process with nonsynonymous mutations that are detected but do not fix included (*Figure 7—figure supplement 1B*). Red squares along the diagonal suggest that the mutations are overdispersed, meaning that nonsynonymous mutations are less likely to fix multiple times in the same population than we would expect by chance. As in *Good et al., 2017*, we quantify this observation of overdispersion by showing that mutations have less 'missed opportunities' than we would expect by chance (*Figure 7—figure supplement 1*).

## Killer phenotype assays

To assay for the killer phenotype, we used a modified halo killing assay (*Woods and Bevan, 1968*). First, we plated a 150 μL of 1:100 diluted saturated culture of a sensitive strain (YAN563) on a single-well (VWR #46600–638) methylene blue agar plate (20 g/L peptone, 10 g/L yeast extract, 20 g/L citric acid monohydrate, 30 mg/L methylene blue, 10 g/L K2HPO4, 20 g/L dextrose, 15 g/L noble agar). We then used the Biomek FXp Liquid Handler to spot 3.5 μL of saturated culture of each of our focal populations onto this lawn. We designed a Biomek protocol that uses a deck spring attachment ('Alpillo' from Alpaqua) to make sure that the tips contacted the agar but did not pierce the agar layer during this step. After 2–3 days of incubation at room temperature, we scanned the plates. We scored each population-timepoint as 'Killer,' 'Diminished Killing,' or 'No Killing,' based on the size of the zone of inhibition (halo) around the spot (see *Figure 9A* for examples of each category, and *Figure 9—figure supplement 1* for the underlying images). The sensitive strain used (YAN563) is a cross between YAN457 (MATa, his3Δ1, ura3Δ0, leu2Δ0, lys2Δ0, RME1pr::ins-308A, ycr043cΔ0::NatMX, can1::RPL39pr_ymGFP_Ste2pr_SpHIS5_Ste3pr_LEU2, derived from BY4742) and YAN433 (MATα, his3Δ1, ura3Δ0, leu2Δ0, lys2Δ0, RME1pr::ins-308A, ycr043cΔ0::HphMX4, can1::RPL39pr_ymCherry_Ste2pr_SpHIS5_Ste3pr_LEU2, derived from BY4742).

## Ploidy assays

To investigate whether any of our focal populations had changed ploidy during the course of the experiment, we measured the DNA content of clones isolated from each focal population at the final timepoint. We isolated one to two clones from each focal population and measured DNA content using a nucleic acid stain as described previously in *Jerison et al., 2020*, but with minor modifications. Briefly, we diluted 4 μL of saturated cultures from each clone (grown in YPD) into 120 μL of water in a 96-well plate, centrifuged the plate, removed the supernatant, resuspended in 50 μL water, added 100 μL of ethanol and pipetted slowly to mix, and incubated at room temperature for 1 hr. Next, we centrifuged the plate, removed the supernatant, let dry for ~5 min, resuspended in 65 μL RNase solution (2 mg/ml RNase in 10 mM Tris-HCl, pH 8.0 and 15 mM NaCl), and incubated at

37°C for 2 hr. We then added 65 µL of 2 µM Sytox Green (Thermo Fisher Scientific S7020), covered the plates in aluminum foil, and shook on a Titramax 100 plate shaker (Heidolph Instruments) for approximately 45 min at room temperature. We measured DNA content using a linear FITC channel on a Fortessa flow cytometer (BD Biosciences). FITC histograms are shown and described in *Figure 8—figure supplement 1*.

## Preliminary imaging

To investigate the possibility of clustering phenotypes in some of our populations with abnormal ploidy stain data, we imaged our focal populations at each of the sequenced timepoints. We diluted cultures 1:360 into 384-well plates (VWR #82051–306) and imaged cells using the Celldiscoverer 7 (Zeiss). Images for all populations are available in *Supplementary file 6*.

## Determining the mutation responsible for a higher mutation rate in *MATa* populations

To investigate the putative higher mutation rate in the *MATa* populations in our experiment, we examined a list of mutations that differentiate our *MATa* ancestor and our *MATα* ancestor (available in *Supplementary file 5*), and identified a putative causative mutation: a missense mutation at a conserved residue in TSA1 (G146S, nucleotide mutation: G->A at bp 436). TSA1 encodes thioredoxin peroxidase, which is involved in eliminating reactive oxygen species that can cause DNA damage, and previous work has shown that deleting the gene causes an increase in mutation rate (*Huang et al., 2003*). As we would expect, this mutation is heterozygous in the diploid ancestor.

We reconstructed the TSA1 G146S mutation by Delitto Perfetto in the S288C background (*Storici and Resnick, 2006*). Briefly, we transformed BY4741 with a KlURA3-KanMX4 cassette, knocking out the whole TSA1 gene, thus creating YAN727. We then removed the cassette with a PCR amplified TSA1 fragment containing the G146S mutation, selecting on 5-FOA and the absence of G418 resistance, thus creating YAN728. The presence of the mutation was then confirmed by Sanger sequencing. We used BY4741 for these reconstructions because our *MATα* ancestor has a functional URA3 (under the STE5 promoter), making it more difficult to create this type of reconstruction.

We performed fluctuation assays on each of the three strains as previously described (*Lang and Murray, 2008*). Briefly, we inoculated a colony from each strain into SC, grew overnight, diluted 1/10,000 and split into 100 µL aliquots in all wells of a 96-well plate, sealed the plate with aluminum foil, and incubated at 30°C without shaking for 48 hr. We combined the eight wells from column 2 of each plate and used the pooled culture to measure cell density on a Coulter Counter Z2 (Beckman Coulter). We spotted the entire volume of each of the other 88 wells on CSM-Arg (Sunrise Scientific) plates supplemented with 100 mg/L L-canavanine (Sigma-Aldrich, St. Louis). Plates were allowed to dry overnight at room temperature, incubated at 30°C for 36 hr, then left at room temperature for 12 hr before counting and scanning. We attempted to count all colonies > 0.25 mm in size in each spotted culture, but note that our estimates of counts > 50 are approximate due to overlapping colonies (counts available in *Supplementary file 5*).

We used the Ma-Sandri-Sarkar Maximum Likelihood Estimator (*Sarkar et al., 1992*), implemented in python (*Bondarev, 2020*) to measure the mutation rate at the CAN1 locus (*Radchenko et al., 2018*; *Sarkar et al., 1992*). As noted in *Lang and Murray, 2008*, there is likely limited postplating growth of sensitive yeast at this Canavanine concentration, leading to the deviations from the expected Luria-Delbruck distribution in *Figure 4—figure supplement 1*. Despite this complication, we can easily see that the TSA1 mutation causes a ~ fivefold increase in mutation rate over the BY4741 background, and the TSA1 deletion causes an ~8-fold increase in mutation rate over the BY4741 background (*Figure 4—figure supplement 1*). In our case, the important result is simply that the TSA1 G146S mutation causes an increase in mutation rate, consistent with the hypothesis that it underlies the higher number of fixed mutations in *MATa* populations in our experiment.

## Data and code accessibility

Raw sequencing data is available on NCBI: https://trace.ncbi.nlm.nih.gov/Traces/sra/?study=SRP286889.

All analysis scripts used in this project are available on GitHub: https://github.com/mjohnson11/VLTE_PIPELINES (copy archived at swh:1:rev:588043a94abb34b13a6dd7a1b25277c25ae8deaf) (archived permanently at https://doi.org/10.5281/zenodo.4422067).

An interactive data browser for this project is available online: https://www.miloswebsite.com/exp_evo_browser.

## Acknowledgements

We thank Andrew Murray, Nina Benites, Yi Chen, members of the Desai lab, members of the Sherlock lab, and three reviewers for helpful comments on the manuscript. We thank the Northwest building staff, in particular Francisco Gonzalez, and the Bauer Core staff, without whom we could not have done this work. This work was supported by NSF Graduate Research Fellowships (MSJ, ERJ, KK, and KRL), the NSF-Simons Center for Mathematical and Statistical Analysis of Biology at Harvard University Grant DMS-1764269 (KRL), the Harvard Program for Research in Science and Engineering (JG), the NDSEG Fellowship Program (CWB), the Fannie and John Hertz Foundation Graduate Fellowship Award (KRL), the Boston Bangalore Biosciences Beginnings Program from DBT, India (RP), the ARC Grant FT 170100441 (MJM), the NSERC (ANNB), Simons Foundation Grant 376196 (MMD), NSF Grant PHY-1914916 (MMD), and NIH Grant R01 GM104239 (MMD). Computational work was performed on the Cannon cluster supported by the Research Computing Group at Harvard University. We thank the Harvard Center for Biological Imaging for infrastructure and support.

## Additional information

### Competing interests

Julia C Piper: Julia C. Piper is affiliated with Aeronaut Brewing Co. The author has no financial interests to declare. The other authors declare that no competing interests exist.

### Funding

| Funder | Grant reference number | Author |
| --- | --- | --- |
| National Science Foundation | Graduate Fellowship | Milo S Johnson<br>Elizabeth R Jerison<br>Katya Kosheleva<br>Katherine R Lawrence |
| Simons Foundation | DMS-1764269 | Katherine R Lawrence |
| Harvard University | PRISE | Juhee Goyal |
| National Defense Science and Engineering Graduate | Graduate Fellowship | Christopher W Bakerlee |
| Hertz Foundation | Graduate Fellowship | Katherine R Lawrence |
| Department of Biotechnology , Ministry of Science and Technology | Boston Bangalore Biosciences Beginnings Program | Ramya Purkanti |
| Australian Research Council | FT170100441 | Michael J McDonald |
| Natural Sciences and Engineering Research Council of Canada | | Alex N Nguyen Ba |
| Simons Foundation | 376196 | Michael M Desai |
| National Science Foundation | PHY-1914916 | Michael M Desai |
| National Institutes of Health | R01 GM104239 | Michael M Desai |

The funders had no role in study design, data collection and interpretation, or the decision to submit the work for publication.

## Author contributions

Milo S Johnson, Conceptualization, Software, Formal analysis, Investigation, Visualization, Methodology, Writing - original draft, Writing - review and editing; Shreyas Gopalakrishnan, Conceptualization, Formal analysis, Investigation, Methodology, Writing - original draft, Writing - review and editing; Juhee Goyal, Formal analysis, Investigation, Methodology; Megan E Dillingham, Investigation, Methodology; Christopher W Bakerlee, Investigation, Writing - review and editing; Parris T Humphrey, Tanush Jagdish, Katherine R Lawrence, Jiseon Min, Alief Moulana, Angela M Phillips, Julia C Piper, Ramya Purkanti, Artur Rego-Costa, Investigation; Elizabeth R Jerison, Katya Kosheleva, Michael J McDonald, Conceptualization, Investigation, Methodology; Alex N Nguyen Ba, Conceptualization, Investigation, Methodology, Writing - review and editing; Michael M Desai, Conceptualization, Formal analysis, Supervision, Funding acquisition, Investigation, Methodology, Writing - original draft, Project administration, Writing - review and editing

## Author ORCIDs

Milo S Johnson https://orcid.org/0000-0003-0169-2494
Shreyas Gopalakrishnan http://orcid.org/0000-0002-7243-0005
Elizabeth R Jerison http://orcid.org/0000-0003-3793-8839
Michael M Desai https://orcid.org/0000-0002-9581-1150

## Decision letter and Author response

Decision letter https://doi.org/10.7554/eLife.63910.sa1
Author response https://doi.org/10.7554/eLife.63910.sa2

# Additional files

## Supplementary files

• Supplementary file 1. Experimental record. Includes daily notes, phenotype information for individual wells (including fitness), and a record of sample sizes for statistical tests.

• Supplementary file 2. A zip file of processed variant calling files for each population.

• Supplementary file 3. A table of all confirmed copy number variants.

• Supplementary file 4. Summary information on mutations, including which genes are mutated in which populations, GO-term enrichments, multi-hit codons, and statistical test results for strain or environment enrichment for each multi-hit gene.

• Supplementary file 5. A record of the detected differences between our ancestral strains and of the fluctuation assay confirming that the TSA1 mutation increases mutation rate.

• Supplementary file 6. A zip file of all preliminary cell imaging.

• Transparent reporting form

## Data availability

Sequencing data have been deposited in the GenBank SRA (accession: SRP286889). Analysis code is available at https://github.com/mjohnson11/VLTE_PIPELINES (copy archived at https://archive.software-heritage.org/swh:1:rev:588043a94abb34b13a6dd7a1b25277c25ae8deaf/). All other generated data is available in Supplementary files 1–6. An interactive data browser is available at https://www.miloswebsite.com/exp_evo_browser.

The following dataset was generated:

| Author(s) | Year | Dataset title | Dataset URL | Database and Identifier |
|---|---|---|---|---|
| Johnson MS, Gopalakrishnan S, Goyal J, Dillingham ME, Bakerlee CW, Humphrey PT, Jagdish T, Jerison ER, Kosheleva K, | 2020 | Evolved *S. cerevisiae* population sequencing | https://trace.ncbi.nlm.nih.gov/Traces/sra/?study=SRP286889 | NCBI Sequencing Read Archive, SRP286889 |

Lawrence KR, Min J, Moulana A, Phillips AM, Piper JC, Purkanti R, Rego-Costa A, McDonald MJ, Ba AN, Desai MM

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
