## [Decision Letter]

All reviewers agreed that your manuscript is suited for *eLife* and, while we do have a few suggestions to further improve the Discussion, none of these are deemed essential. Hence, we are pleased to inform you that your article, "Phenotypic and molecular evolution across 10,000 generations in laboratory budding yeast populations", has been accepted for publication in *eLife*.

We appreciate how your expensive study contributes to our understanding of experimental evolution by providing one of the longer running experiments, here using a eukaryote (*Saccharomyces cerevisiae*). The comparison of your data to those obtained from the famous long-term evolution experiment in *E. coli* as well as many other shorter-term experiments in yeast does not only confirm some of the major findings regarding the molecular processes that drive evolution, but also identifies a few interesting differences, most notably regarding the declining effect fitness effect of beneficial mutations, the importance of mutators strains, and the role of polyploidization.

Although we do not think that any of the comments that the reviewers made were truly essential, we do strongly encourage you to carefully consider the suggestions and questions, particularly those that revolve around the differences with the LTE experiment in E coli. We have therefore copied the integral reviews below.

Reviewer #1:

This study tracks the evolution of 205 *S. cerevisiae* populations over roughly 10,000 generations. In brief, the authors confirm observations emerging from previous, similar experiments in yeast and *E. coli*, with populations showing a declining increase in fitness over time and various complex interactions between sub-populations in the same population, as well as between mutations in the same genome.

While some of the findings reported in this particularly eloquent manuscript are perhaps not completely novel, the study distinguishes itself from previous work by its vast scale (number of populations and generations, as well as comparing haploids vs. diploids) and the comprehensive analyses, which together yield a more integrative view on the different evolutionary forces and mechanisms at play. As such, I believe that the report furthers our understanding of evolution, and perhaps in particular domestication.

One major difference with the LTE experiment in *E. coli* is the size of the populations. Would this explain some of the observed differences? I believe this could use a bit more discussion.

The different adaptive dynamics and patterns in haploids and diploids is perhaps the most interesting and most novel part of the study, and I believe that this merits more attention in the Discussion section. The authors carefully consider different possible reasons for the slower adaptation observed in diploids, as well as the potential role of LOH and sex (including inbreeding). The idea that inbreeding could help purge deleterious heterozygous hitchhiker mutations is particularly interesting and might even explain why yeast cells have evolved the mating type switching system that promotes inbreeding (apart from the fact that it also makes it easier for cells to escape the haploid form after a sporulation event, which can be crucial to survive famine and other stressful environments). I wonder whether the authors can comment a bit more on these aspects. Moreover, the quick and likely irreversible loss of the capacity to go through a sexual cycle may also tell us that lab evolution experiments are very far removed from evolution in nature, since most wild strains show efficient meiosis. Domesticated industrial yeasts, on the other hand, show very similar patterns of evolution, suggestion that domestication of industrial microbes is in fact very similar to the directed evolution experiments carried out in labs (e.g. my lab's own work: DOI:https://doi.org/10.1016/j.cell.2016.08.020 )

Discussion: The rise of mutator phenotypes might also depend on the levels and nature of selective pressure imposed on the cells? Conditions that severely hamper growth (and thus the supply of mutations), or stress conditions that require the combined effect of two (or more) mutations may be more prone to drive the evolutionary success of mutators? See for example DOI: 10.7554/*eLife*.22939.

Reviewer #2:

I read with interest the manuscript "Phenotypic and molecular evolution across 10,000 generations in laboratory budding yeast populations". It adds to our understanding of experimental evolution by providing one of the longer running experiments, here using a eukaryote (*Saccharomyces cerevisiae*). As the authors note, previous experiments of a similar length (>=10,000 generations) have been conducted with bacteria (especially the LTEE experiment of Lenski and colleagues).

The manuscript is generally well written, and I believe deserving of a major publication with broad readership. A few larger suggestions are to discuss more thoroughly the following points:

1) One of the biggest puzzles in my mind is why the authors didn't see *more* polyploidization. As noted by Harari et al., 2018, and found in many previous long-running experiments (starting with Gerstein et al., 2006, and many others noted in Table 1 of Fisher et al., 2018), spontaneous diploidization is common in experimental evolution studies with yeast. Given that there was such a large initial fitness advantage to diploids (Figure 2), the fact that only one line diploidized over a much longer time span is interesting. Is this something peculiar to the strain or the environmental conditions (I didn't see anything that would have selectively maintained haploids in the Materials and methods, although the lines were constructed to be able to selectively maintain haploids versus diploids). Harari et al. studied several strains, so it isn't obvious that the W303-derived strain used here would differ.

2) A main result is the diminishing nature of the fitness gains. This is certainly not surprising, and I believe the result, but I was surprised that this was not quantitatively tackled. One of the biggest questions addressed from the LTEE is whether fitness approaches an asymptote or continues to increase at a slowing rate. It is almost certainly too early to address this question definitively, but fitting different functions and noting what can or cannot be rejected at this stage would be a welcome addition to the paper.

3) Similarly, it is claimed that the variability among lines decreases (subsection “Fitness changes during evolution”), but this is neither statistically confirmed nor visually obvious in Figure 2.

4) When discussing the lack of mutators, perhaps also note the difference in ancestral mutation rates. *S. cerevisiae* has a larger genome-wide mutation rate than *E. coli* (Lynch et al., 2016), which would have decreased the advantage to rare mutators.

5) Another major difference from the LTEE is said to be the lack of coexisting strains. I'm not convinced that this is a qualitative difference. Coexistence is only possible once the period of strongly adaptive sweeps is over (i.e., where the fitness benefit of any adaptive mutation falls below the fitness difference between the coexisting strains). Given the strong signs of adaptive mutations seen in Figure 3 (including the supplement), it could well be that there are plenty of mutations that could coexist, but that hitchhiking has eliminated them. Only once fitness gains have really slowed might they be seen to persist.

6) It is claimed in the Discussion at the authors find only "one strong case of repeatability at the level of the nucleotide change", but this hasn't actually been discussed in the text. Which case are they referring to? There are several listed Figure 5—figure supplement 1. Also, the reader cannot tell whether contamination is a possible source of the repeated hits in this supplementary figure. Please discuss.

Other comments:

1) The first page of the Introduction is too generic and could be cut or shortened.

2) The claim made in the text that "We find a different pattern in some diploid populations, where an initial slower period of fitness gain is succeeded by a significant rapid increase in fitness" is not supported by the data presented. I would move Figure 2—figure supplement 1B to Figure 2 as an second row of panels. Even then, as presented in these additional panels, the diploids only appear to adapt rapidly later on in SC30; this might be because of the course division into halves of the experiment in Figure 2—figure supplement 1B (would thirds be clearer?), but at any rate the statistical evidence for this claim needs to be added.

3) Related to the above point, one of the main explanations for why diploid adaptation is expected to speed up is if the available beneficial mutations are partially recessive, giving a small boost when in heterozygotes but only a large boost in fitness once an LOH event occurs. This is of interest, but it requires showing statistically that there is an acceleration in fitness gain in the diploids, which is not clear.

4) Introduction and subsection “Patterns of molecular evolution specific to diploids” – A direct comparison of haploid and diploid rates of adaptation was conducted by Gerstein et al., 2011, [The Marad experiment is good to cite, but the haploid and diploid experiments weren't contemporaneous and only in one environment. The Fisher et al. experiment isn't really as relevant to this point but is very relevant to the routes of adaptation accessible to haploids versus diploids and the difference in autodiploidization rates.]

5) "agreement with this hypothesis" – The finding of slow then fast adaptation is not in agreement with the hypothesis in the previous sentence that diploids adapted more slowly because they started at a higher fitness. That hypothesis does not explain why diploids would speed up.

6) Subsection “Molecular evolution” and elsewhere – In all comparisons with results from previous results (especially LTEE), please provide the generation at which a comparison can be made (ideally at 10,000). Results from LTEE that didn't appear until much later aren't reliably a difference from the current experiment. This is touched upon in the Discussion but is needed at first mention.

References:

Gerstein, A. C., Chun, H. J. E., Grant, A., and Otto, S. P. (2006). Genomic convergence toward diploidy in *Saccharomyces cerevisiae*. PLoS genetics, 2(9), e145.

Gerstein, A. C., Cleathero, L. A., Mandegar, M. A., and Otto, S. P. (2011). Haploids adapt faster than diploids across a range of environments. Journal of evolutionary biology, 24(3), 531-540.

Harari, Y., Ram, Y., Rappoport, N., Hadany, L., and Kupiec, M. (2018). Spontaneous changes in ploidy are common in yeast. Current Biology, 28(6), 825-835.

Lynch, M., Ackerman, M. S., Gout, J. F., Long, H., Sung, W., Thomas, W. K., and Foster, P. L. (2016). Genetic drift, selection and the evolution of the mutation rate. Nature Reviews Genetics, 17(11), 704.

Reviewer #3:

1) Please make explicit how generations are calculated. Is this estimated number of generations per week on each media updated with the empirical results of fitness assays? Or simply assumed to remain constant?

2) Please clarify the rationale for varying the reference strain for calculating fitness among environments in the text. (Presumably, a concern about sensitivity for fitness assays at different absolute fitness.)

3) In competitive fitness assays, it is more ideal to have both strains marked with a fluorescent marker for the reason the authors highlight-fluorescent genotypes will typically produce a small non-fluorescent subset in the population-but given the design here, running the assay with one marked strain is fine. I miss some detail in the supplement about the nature of the fitness calculations, and it does not appear the authors have provided the raw ratios of the flow cytometry data used to calculate fitness. The availability of the code for analysis of this data is noted in the Transparent Reporting document, but it would be helpful to make the information more prominent in the manuscript text.

4) Relatedly, in Figure 2—figure supplement 2, there appear to be a set of strains in the SC 37°C environment that showed low fitness in replicate 2 compared to replicate 1. This is not necessarily problematic given the scale of the analysis presented here, but as written the manuscript is somewhat opaque about exactly what technical (or biological) variation could contribute to this pattern. Please clarify by providing the raw fitness measurements and/or by addressing this small discrepancy directly.

5) It might be helpful to readers to annotate the pathway in Figure 7A with the frequency of mutations in each gene.

6) The majority of the beneficial *ade2* stop codon reversions observed are in diploids. The authors make the case for target size of the stop codon reversion (vs. hitting of other parts of the pathway) making a difference to the frequency with which populations find the functional ADE fitness peak. Is a difference in target size for haploids and diploids relevant here?

7) It would be helpful provide a brief description of the mutual information inference for readers.

---

## [Author Response]

Reviewer #1:[…] One major difference with the LTE experiment in *E. coli* is the size of the populations. Would this explain some of the observed differences? I believe this could use a bit more discussion.

Our populations are considerably smaller than the LTEE populations, and this could certainly be part of the reason we don’t see as much long-term coexistence as in the LTEE. We added a sentence to the Discussion about this:

“Alternately, coexisting lineages may have emerged in our experiment but been lost due to strong within-lineage adaptation or drift. Since our populations are smaller than those in the LTEE, low-frequency lineages will be lost more commonly during the daily bottleneck.”

And we added a sentence to the Materials and methods detailing population sizes:

“These dilutions determine the number of doublings or generations per day (10 for YPD 30°C and SC 30°C, 8 for SC 37°C), the bottleneck population size (~8 • 10^3^ for YPD 30°C and SC 37°C, ~2 • 10^3^ for SC 30°C), and the corresponding effective population size (~6 • 10^4^ for YPD 30°C, ~4 • 10^4^ for SC 37°C, ~1 • 10^4^ for SC 30°C) (Wahl and Gerrish, 2001). These bottleneck sizes are based on estimated saturation densities of ~6 • 10^7^ for YPD and ~1.6 • 10^7^ for SC in 96-well plates, measured using a Coulter Counter Z2 (Beckman Coulter).”

The different adaptive dynamics and patterns in haploids and diploids is perhaps the most interesting and most novel part of the study, and I believe that this merits more attention in the Discussion section. The authors carefully consider different possible reasons for the slower adaptation observed in diploids, as well as the potential role of LOH and sex (including inbreeding). The idea that inbreeding could help purge deleterious heterozygous hitchhiker mutations is particularly interesting and might even explain why yeast cells have evolved the mating type switching system that promotes inbreeding (apart from the fact that it also makes it easier for cells to escape the haploid form after a sporulation event, which can be crucial to survive famine and other stressful environments). I wonder whether the authors can comment a bit more on these aspects. Moreover, the quick and likely irreversible loss of the capacity to go through a sexual cycle may also tell us that lab evolution experiments are very far removed from evolution in nature, since most wild strains show efficient meiosis. Domesticated industrial yeasts, on the other hand, show very similar patterns of evolution, suggestion that domestication of industrial microbes is in fact very similar to the directed evolution experiments carried out in labs (e.g. my lab's own work: DOI:https://doi.org/10.1016/j.cell.2016.08.020)

Thanks for bring this to our attention. It is particularly interesting to think about whether the accumulation of deleterious recessive mutations during a period of asexual (or mostly asexual) evolution in these domesticated strains could lead to low spore viability and then lead to relaxed selection on phenotypes needed for the sexual cycle. Of course this is a bit of a chicken-or-the-egg problem! It would be interesting to see if signatures of loss of heterozygosity are different in these industrial strains that appear to be completely asexual, although we’re not sure to what extent it is possible to distinguish between LOH and recombination from mating in these cases. For the purposes of this paper, we have added a line to the Results paragraph where we discuss this phenomenon to draw attention to this connection to domestication (new text is underlined):

“Thus, passage through Haldane’s sieve by loss of heterozygosity should become less likely as populations accumulate a substantial load of hitchhiking heterozygous mutations. […] However, we note that in sexual lineages recombination with sufficient inbreeding could dramatically alter these dynamics, by continuously purging recessive deleterious load (Charlesworth and Willis, 2009).”

Discussion: The rise of mutator phenotypes might also depend on the levels and nature of selective pressure imposed on the cells? Conditions that severely hamper growth (and thus the supply of mutations), or stress conditions that require the combined effect of two (or more) mutations may be more prone to drive the evolutionary success of mutators? See for example DOI: 10.7554/eLife.22939.

We agree that differences in the environment and the ancestral genotypes may contribute to changes in the distribution of fitness effects that could change indirect selection for mutators. We added a parenthetical to the sentence where we discuss this to specifically highlight the potential role of environmental conditions or differences in ancestral fitness, with citations of examples:

“Instead, the lack of mutator lineages may stem from a difference in the rate at which mutators arise or a different balance between the relative importance of beneficial and deleterious mutations (which depends on the environment and ancestral fitness; see e.g. Swings et al., 2017 and Kryazhimskiy et al., 2014) that leads to less indirect selection for mutators (Good and Desai, 2016).”

Reviewer #2:[…] The manuscript is generally well written, and I believe deserving of a major publication with broad readership. A few larger suggestions are to discuss more thoroughly the following points:1) One of the biggest puzzles in my mind is why the authors didn't see more polyploidization. As noted by Harari et al., 2018, and found in many previous long-running experiments (starting with Gerstein et al., 2006, and many others noted in Table 1 of Fisher et al., 2018), spontaneous diploidization is common in experimental evolution studies with yeast. Given that there was such a large initial fitness advantage to diploids (Figure 2), the fact that only one line diploidized over a much longer time span is interesting. Is this something peculiar to the strain or the environmental conditions (I didn't see anything that would have selectively maintained haploids in the Materials and methods, although the lines were constructed to be able to selectively maintain haploids versus diploids). Harari et al. studied several strains, so it isn't obvious that the W303-derived strain used here would differ.

The first thing we did when starting to gather this data was a preliminary ploidy stain and we were also surprised to find low levels of polyploidization. While there are many possible explanations involving differences in selection pressure between haploids and diploids, we believe this is largely due to a lower rate of endoreduplication in our strains. Contrary to Fisher et al., 2018, we have found that autodiploidization is quite rare in our W303 strains. The reason for this discrepancy remains somewhat of a mystery to us, but we have a paper in preparation identifying the genetic basis for this difference between W303 and other strains. Data analysis for that paper is still in progress, but we have added a sentence to more accurately reflect how surprising this result was, citing more previous work:

“We were surprised by this result; previous studies have demonstrated that evolving haploid yeast populations often become diploid (Gerstein et al., 2006; Fisher et al., 2018; Harari, 2018).”

2) A main result is the diminishing nature of the fitness gains. This is certainly not surprising, and I believe the result, but I was surprised that this was not quantitatively tackled. One of the biggest questions addressed from the LTEE is whether fitness approaches an asymptote or continues to increase at a slowing rate. It is almost certainly too early to address this question definitively, but fitting different functions and noting what can or cannot be rejected at this stage would be a welcome addition to the paper.

We appreciate this point, and agree it should be a part of future analysis once more generations have elapsed and more temporally dense fitness assays are completed (possibly on a smaller set of populations). Unfortunately, we don't think our current data isn't suited towards this kind of analysis. We have temporally sparse fitness measurements over the second half of evolution compared to Wiser et al., 2013, and we observe some fitness assay batch effects (discussed in the Materials and methods) that, while small on the overall scale of fitness change over the course of the experiment, could have a significant (and spurious) effect on this kind of model fitting.

3) Similarly, it is claimed that the variability among lines decreases (subsection “Fitness changes during evolution”), but this is neither statistically confirmed nor visually obvious in Figure 2.

Thanks for bringing this to our attention – upon more close analysis, this is not universally true in our experiment: the variability between all evolving populations does decrease in YPD 30°C and SC 37°C, but it increases in SC 30°C, driven by low-fitness diploids and high-fitness *ade2-1*-reversion lines. We have removed this sentence.

4) When discussing the lack of mutators, perhaps also note the difference in ancestral mutation rates. *S. cerevisiae* has a larger genome-wide mutation rate than *E. coli* (Lynch et al., 2016), which would have decreased the advantage to rare mutators.

Thanks for bringing this up – it seems likely to us that this could be part of the reason for a lack of mutators, and it was missing from the Discussion. We have added this sentence to the Discussion:

“We also may have less second-order selection for mutators in our experiment because our strains have mutation rates that are higher than the ancestral *E. coli* strain in the LTEE, though lower than the LTEE mutator lineages (Lang and Murray, 2008; Wielgoss et al., 2013).”

5) Another major difference from the LTEE is said to be the lack of coexisting strains. I'm not convinced that this is a qualitative difference. Coexistence is only possible once the period of strongly adaptive sweeps is over (i.e., where the fitness benefit of any adaptive mutation falls below the fitness difference between the coexisting strains). Given the strong signs of adaptive mutations seen in Figure 3 (including the supplement), it could well be that there are plenty of mutations that could coexist, but that hitchhiking has eliminated them. Only once fitness gains have really slowed might they be seen to persist.

We appreciate this point, and think that it is related to a comment from another reviewer about differences in population size between our populations and LTEE populations. Mutations sweeping within coexisting lineages may shift the equilibrium frequency to a low/high frequency, such that one lineage is easily lost by drift at the bottleneck. We’re not sure how much this effect depends on the stage of adaptation, though as you note we would naively expect it to be strongest early in evolution; however, in looking back at the coexisting lineages described in Good et al., 2017, coexisting lineages change equilibrium frequency by large amounts even late in the LTEE. In any case, we agree that the possibility that within-lineage adaptation caused coexisting lineages to be eliminated is an important point that was missing from our Discussion. We have added this sentence to the Discussion, and have reported approximate population sizes in the Materials and methods (see reviewer 1’s first comment above):

“Alternately, coexisting lineages may have emerged in our experiment but been lost due to drift or strong within-lineage adaptation. Since our populations are smaller than those in the LTEE, low-frequency lineages will be lost more commonly during the daily bottleneck.”

6) It is claimed in the Discussion at the authors find only "one strong case of repeatability at the level of the nucleotide change", but this hasn't actually been discussed in the text. Which case are they referring to? There are several listed Figure 5—figure supplement 1. Also, the reader cannot tell whether contamination is a possible source of the repeated hits in this supplementary figure. Please discuss.

We are referring to the *ade2-1* reversions here, which we have now directly referenced in the text, and we have also clarified that this is the only strong case for repeated *selection* at the level of nucleotide change, since the other changes listed in Figure 5—figure supplement 1 are likely to be the result of frequent mutation in repetitive regions or (in one case) chance (note that Figure 5—figure supplement 1 is now included in Supplementary file 4 because *eLife* does not allow tables as figure supplements). The text now reads:

“We find only one strong case of repeatable selection at the level of the nucleotide change (*ade2-1* reversions)”.

The evidence that these mutations are not the result of cross-contamination is in the allele-frequency data: each population has unique mutations that are fixed early in the experiment and remain fixed throughout. We have also investigated all of these cases individually in the interactive data browser to ensure they are not alignment errors or sequencing contamination artifacts.

Other comments:1) The first page of the Introduction is too generic and could be cut or shortened.

We appreciate this feedback but have decided to leave the Introduction in its broad format in this revision, which we hope is interesting to readers not directly familiar with the field.

2) The claim made in the text that "We find a different pattern in some diploid populations, where an initial slower period of fitness gain is succeeded by a significant rapid increase in fitness" is not supported by the data presented. I would move Figure 2—figure supplement 1B to Figure 2 as an second row of panels. Even then, as presented in these additional panels, the diploids only appear to adapt rapidly later on in SC30; this might be because of the course division into halves of the experiment in Figure 2—figure supplement 1B (would thirds be clearer?), but at any rate the statistical evidence for this claim needs to be added.

We agree that this was too broadly worded, without convincing evidence that this is a systematic pattern. We have amended the clause to read "some diploid populations in SC 30°C" and have cut the following sentence about diploid populations "catching up" in the second half of the evolution. We appreciate the suggestion to move Figure 2—figure supplement 1B to the main figure, but have decided to keep only the raw fitness data in Figure 2 because we believe the main point of Figure 2—figure supplement 1 (to show declining adaptability) is qualitatively clear in the existing panels of Figure 2.

3) Related to the above point, one of the main explanations for why diploid adaptation is expected to speed up is if the available beneficial mutations are partially recessive, giving a small boost when in heterozygotes but only a large boost in fitness once an LOH event occurs. This is of interest, but it requires showing statistically that there is an acceleration in fitness gain in the diploids, which is not clear.

We agree that this effect may be playing an important role. Our best guess is that either this effect, or a related effect in which beneficial mutations are fully recessive and populations must wait for LOH events for selection to act on these mutations at all, is responsible for the rapid fitness increases in diploid populations in SC 30°C later in the experiment. As you say, however, to demonstrate these effects we would need to show acceleration in fitness gains in these populations, ideally concurrent with LOH events. While we can anecdotally find cases in which large fitness increases in diploid populations in SC 30°C are concurrent with homozygous fixation of mutations in the ADE pathway (the easiest way to see this is using the interactive data browser, hosted at https://www.miloswebsite.com/exp_evo_browser), we don't have the statistical power to demonstrate that these cases are responsible for broad patterns in fitness gains.

4) Introduction and subsection “Patterns of molecular evolution specific to diploids” – A direct comparison of haploid and diploid rates of adaptation was conducted by Gerstein et al., 2011, [The Marad experiment is good to cite, but the haploid and diploid experiments weren't contemporaneous and only in one environment. The Fisher et al. experiment isn't really as relevant to this point but is very relevant to the routes of adaptation accessible to haploids versus diploids and the difference in autodiploidization rates.]

Thanks for bringing this to our attention! We have added citations to Gerstein et al. 2011, at multiple sites in the text and agree that this is a much more comprehensive and strong demonstration of the differences in rates of adaptation between haploids and diploids.

5) "agreement with this hypothesis" – The finding of slow then fast adaptation is not in agreement with the hypothesis in the previous sentence that diploids adapted more slowly because they started at a higher fitness. That hypothesis does not explain why diploids would speed up.

Yes, you are correct! We originally included this thought since the observation of diploids adapting faster than haploids during the second half of the experiment in that environment seemed striking, and different from previous results – but upon considering it more, this effect is likely due to the fact that some of these diploid populations do not have an ADE pathway mutation fixed at the start of the second half of evolution, meaning that the increased rate of adaptation in diploids is likely due to a difference between the genotypes the haploid and diploid populations begin the second half of evolution with. We have removed this sentence along with the final sentence of the paragraph, and we think the shortened paragraph now presents a more balanced Discussion: our data are consistent with previous data (including Gerstein et al., 2011) showing faster adaptation in diploids, which may be entirely due to the effects of dominance, but may also be partially due to widespread patterns of declining adaptability caused by global diminishing returns epistasis. We have also tried to balance the wording in this sentence, which appears later in the text:

“Thus the slower rate of fitness increase in diploids could instead also partly be or entirely a consequence of diminishing returns (Chou et al., 2011; Khan et al., 2011; Kryazhimskiy et al., 2014).”

6) Subsection “Molecular evolution” and elsewhere – In all comparisons with results from previous results (especially LTEE), please provide the generation at which a comparison can be made (ideally at 10,000). Results from LTEE that didn't appear until much later aren't reliably a difference from the current experiment. This is touched upon in the Discussion but is needed at first mention.References:Gerstein, A. C., Chun, H. J. E., Grant, A., and Otto, S. P. (2006). Genomic convergence toward diploidy in Saccharomyces cerevisiae. PLoS genetics, 2(9), e145.Gerstein, A. C., Cleathero, L. A., Mandegar, M. A., and Otto, S. P. (2011). Haploids adapt faster than diploids across a range of environments. Journal of evolutionary biology, 24(3), 531-540.Harari, Y., Ram, Y., Rappoport, N., Hadany, L., and Kupiec, M. (2018). Spontaneous changes in ploidy are common in yeast. Current Biology, 28(6), 825-835.Lynch, M., Ackerman, M. S., Gout, J. F., Long, H., Sung, W., Thomas, W. K., and Foster, P. L. (2016). Genetic drift, selection and the evolution of the mutation rate. Nature Reviews Genetics, 17(11), 704.

All comparisons made with the LTEE are made at the 10,000 generation mark, and we have added phrases and references to the main text to reflect this for each comparison.

Reviewer #3:1) Please make explicit how generations are calculated. Is this estimated number of generations per week on each media updated with the empirical results of fitness assays? Or simply assumed to remain constant?

We have added a reference to the fact that the dilution factor determines generation numbers: “These dilutions determine the number of doublings or generations per day (10 for YPD 30°C and SC 30°C, 8 for SC 37°C)”.

2) Please clarify the rationale for varying the reference strain for calculating fitness among environments in the text. (Presumably, a concern about sensitivity for fitness assays at different absolute fitness.)

We explain this decision in the Materials and methods: “In order to assess competitive fitness using a consistent reference for each environment, we isolated clones at various generations from an arbitrarily chosen evolving diploid population in YPD 30°C (P1G09). We looked for clones that had fitnesses intermediate between the ancestral strains and evolved strains in each environment, and tagged these clones by inserting a yNatMX cassette…”

3) In competitive fitness assays, it is more ideal to have both strains marked with a fluorescent marker for the reason the authors highlight-fluorescent genotypes will typically produce a small non-fluorescent subset in the population-but given the design here, running the assay with one marked strain is fine. I miss some detail in the supplement about the nature of the fitness calculations, and it does not appear the authors have provided the raw ratios of the flow cytometry data used to calculate fitness. The availability of the code for analysis of this data is noted in the Transparent Reporting document, but it would be helpful to make the information more prominent in the manuscript text.

We have now included raw counts for cells in the reference gates, along with the calculated uncorrected and corrected reference frequencies in each sample in Supplementary file 1, in the "Fitness Assay Counts" sheet, and we note that interested readers can look at graphs of the raw flourescence and gating in the interactive browser, hosted at https://www.miloswebsite.com/exp_evo_browser.

4) Relatedly, in Figure 2—figure supplement 2, there appear to be a set of strains in the SC 37°C environment that showed low fitness in replicate 2 compared to replicate 1. This is not necessarily problematic given the scale of the analysis presented here, but as written the manuscript is somewhat opaque about exactly what technical (or biological) variation could contribute to this pattern. Please clarify by providing the raw fitness measurements and/or by addressing this small discrepancy directly.

We have provided an explanation for this deviation from the 1:1 replicate correlation line in the Materials and methods text. We believe the basic issue is that minor batch effects, likely caused by the two different machines used for the two replicates, have an outsized effect when reference frequencies rise quickly (in low-fitness populations). The Materials and methods text added reads:

“While our replicate fitness measurements generally correlate very well (Figure 2—figure supplement 1), there is a deviation from the 1:1 line for replicate fitness measurements for low-fitness populations in SC 37°C. […] Only assays in which the reference frequency goes up very rapidly (low-fitness populations) are strongly affected.”

5) It might be helpful to readers to annotate the pathway in Figure 7A with the frequency of mutations in each gene.

Thanks for this suggestion, we have made this change.

6) The majority of the beneficial ade2 stop codon reversions observed are in diploids. The authors make the case for target size of the stop codon reversion (vs. hitting of other parts of the pathway) making a difference to the frequency with which populations find the functional ADE fitness peak. Is a difference in target size for haploids and diploids relevant here?

We think this comment is referring to the fact that diploids and haploids have two and one copies of ADE2, respectively. This difference would matter in the case that the process of finding the functional ADE fitness peak was fully mutation-limited, but the important dynamic here – as the reviewer notes – is the relative rate of transition to the functional ADE fitness peak or the lower-fitness state associated with mutations upstream in the pathway. Therefore, we think the key distinction between haploids and diploids in terms of evolutionary outcome is the dominance of mutations in the two pathways. This distinction might be possible to frame as an issue of “target size” if we use that term in a broad sense that includes the rate of loss of heterozygosity in diploids: in that case the target sizes of the two evolutionary paths are more similar to each other in diploids than in haploids. However, we think this broad use of target size would be confusing to many readers, so we have decided to leave our original discussion, which frames the difference in terms of dominance and Haldane’s sieve.

7) It would be helpful provide a brief description of the mutual information inference for readers.

We have added a more extensive description of the mutual information inference, which directly follows Fisher et al., 2019. See “Mutual Information Analysis” subsection in the revised text. Note that since we reran simulations associated with our analysis, the statistical results for this analysis, along with those for our overdispersion analysis, are very slightly different from the original draft (but no results have changed).

Lastly, we’d like to note that we made a few minor changes based on feedback from colleagues who read the preprint. Notably, we added *rad5-535* to the reported strain genotype and fixed a record-keeping mistake in which our generations for the SC 37°C were off by 2 generations.